

# Measurement Report: Online Measurement of Gas-Phase Nitrated Phenols Utilizing CI-LToF-MS: Primary Sources and Secondary Formation

Kai Song[1,2], Song Guo[1,2*], Haichao Wang[3], Ying Yu[1], Hui Wang[1], Rongzhi Tang[1], Shiyong Xia[4], Yuanzheng Gong[1], Zichao Wan[1], Daqi Lv[1], Rui Tan[1], Wenfei Zhu[1], Ruizhe Shen[1], Xin Li[1], Xuena Yu[1], Shiyi Chen[1], Liming Zeng[1], Xiaofeng Huang[4]

[1]State Key Joint Laboratory of Environmental Simulation and Pollution Control, International Joint Laboratory for Regional Pollution Control, Ministry of Education (IJRC), College of Environmental Sciences and Engineering, Beijing, 100871, China
[2]Collaborative Innovation Center of Atmospheric Environment and Equipment Technology, Nanjing University of Information Science & Technology, Nanjing 210044, China
[3]School of Atmospheric Sciences, Sun Yat-sen University, Zhuhai, 519082, China
[4]Key Laboratory for Urban Habitat Environmental Science and Technology, School of Environment and Energy, Peking University Shenzhen Graduate School, Shenzhen, 518055, China

*Correspondence to*: Song Guo (songguo@pku.edu.cn)

**Abstract.** To investigate the composition, variation, and sources of nitrated phenols (NPs) in the winter of Beijing, gas-phase NPs were measured by using a chemical ionization long time-of-flight mass spectrometer (CI-LToF-MS). A box model was applied to simulate the secondary formation process of NPs. In addition, the primary sources of NPs were resolved by non-negative matrix factorization (NMF) model. Our results showed that secondary formation contributed 38%, 9%, 5%, 17% and almost 100% of the ambient nitrophenol (NP), methyl-nitrophenol (MNP), dinitrophenol (DNP), methyl-dinitrophenol (MDNP or DNOC), and dimethyl-nitrophenol (DMNP). Phenol-OH reaction was the predominant loss pathway (46.7%) during the heavy pollution episode, which produced phenoxy radical ($C_6H_5O$). The phenoxy radical consequently reacted with $NO_2$, and produced nitrophenol. By estimating the primarily emitted phenol from the ratio of phenol/CO from freshly emitted vehicle exhaust, this study proposed that oxidation of primary phenol contributes much more nitrophenol (37%) than that from benzene oxidation (<1%) in the winter of Beijing. The latter pathway was widely used in models and might lead to great uncertainties. The source apportionment results by NMF indicated the importance of combustion sources (>50%) to the gas-phase NPs. The industry source contributed 30% and 9% to DNP and MDNP, respectively, which was non-negligible. The concentration weighted trajectory (CWT) analysis demonstrated that regional transport from provinces that surround the Yellow and Bohai Seas contributed more primary NPs to Beijing. Both primary sources and secondary formation in either local or regional scale should be considered when making NPs control policies.



## 1 Introduction

Nitrated phenols (NPs) refer to aromatic compounds with at least a hydroxyl (-OH) group and a nitro (-NO$_2$) group. They have gained much concern due to forest decline and phytotoxic activities (Grosjean and Williams, 1992; Qingguo Huang et

al., 1995). Besides, NPs are important components of brown carbon with absorption properties in near UV-light (Iinuma et al., 2010; Laskin et al., 2015; Lu et al., 2019a; Xie et al., 2017). As a result, NPs were widely detected around the world in the gas and particle phase, in fog, cloud, rain, snow and surface water since the 1980s (Harrison et al., 2005). Among these studies, gas-phase NPs were detected in urban, suburban, and remote regions (Mohr et al., 2013; Morville et al., 2006; Priestley et al., 2018). The concentration of NPs varied significantly from place to place (Harrison et al., 2005). Beijing was

the capital city of China which retains a population of more than 20 million and preserves more than 5 million private cars, yet the occurrence of gas-phase NPs in Beijing was rarely investigated. Most of the studies in Beijing focus on particle-phase NPs (or NACs) (Li et al., 2020; Wang et al., 2019b). The estimated gas-phase concentration of nitrophenol from particle-phase was as much as 600 ppt without direct evidence of measurement (Wang et al., 2019b). Consequently, it is of vital importance to identify the concentration and sources of NPs in Beijing.

Gas chromatography-mass spectrometer (GC-MS) and high-performance liquid chromatography-mass spectrometer (HPLC-MS) were commonly used to quantify the ambient concentration of NPs with accurate molecular information (Belloli et al., 1999; Harrison et al., 2005; Lüttke et al., 1997). Conversely, the pretreatment procedure is frustrating and the time resolution is rather low. The measurement of reactive atmospheric phenolic compounds demands a real-time, high time resolution and accurate method. In recent years, chemical ionization mass spectrometry (CIMS) has become popular for its high accuracy

and time resolution (<1s) (Priestley et al., 2018; Yatavelli et al., 2012). The oxidation routines of different organic compounds have been clarified by the online approach of CIMS (Bannan et al., 2015; Mohr et al., 2013; Yuan et al., 2016; Zheng et al., 2015). Accordingly, CIMS is a powerful approach in measuring atmospheric organic compounds, which is appropriate for the quantification of ambient NPs.

NPs in the atmosphere come from both primary emission and secondary formation. Coal combustion, biomass burning and

vehicle exhaust are the common sources of primary NPs emission (Lu et al., 2019a, 2019b; Wang et al., 2017). Besides, phenolate compounds are widely used as drugs, plastics and antioxidants (Heberer and Stan, 1997). Dinitrophenol (DNP) and methyl-dinitrophenol (MDNP, or known as DNOC) have been used as pesticides for more than 50 years (Chaara et al., 2010). As a result, the pesticide industry may be a probable source of DNP and MDNP emissions. Despite the complex primary emissions, the secondary formation of atmospheric NPs is also crucial (Harrison et al., 2005; Yuan et al., 2016).

Photooxidation of benzene, toluene and xylene by OH and NO$_3$ forms phenol, cresol and xylenol, respectively. The further oxidation of these phenols results in the secondary formation of nitrophenol (NP), methyl-nitrophenol (MNP) and dimethyl-nitrophenol (DMNP) (Harrison et al., 2005). However, not only do the atmospheric phenols come from the oxidation of aromatics but also they are emitted directly from vehicle exhaust, biomass burning and other primary sources (Inomata et al., 2014; Laskin et al., 2015; Sekimoto et al., 2013).To make more useful NPs control strategies, it is of vital importance to



distinguish the proportion of secondary formation of NPs from benzene and that from the oxidation of the directly emitted phenols.

In the present work, we conducted high time resolution measurement of the gas-phase nitrated phenols by using a chemical ionization long time-of-flight mass spectrometer (CI-LToF-MS, CIMS) in the winter of Beijing. The secondary formation process of NPs was simulated by a box model. The primary phenol oxidation process was distinguished from benzene

oxidation to investigate its role in the secondary formation of NPs. Non-negative matrix factorization (NMF) and concentration weighted trajectory (CWT) analysis were utilized to construct the source apportionment and identify the potential region of these sources.

## 2  Materials and Methods

### 2.1  Measurements of nitrated phenols and other gaseous pollutants

#### 2.1.1  Measurement location

The sampling site is at an urban site, i.e. Peking University Atmosphere Environment MonitoRing Station (PKUERS, 39° 59′ N, 116°18′ E), which is located on the campus of Peking University. The details about this site were reported in the previous work(Guo et al., 2012, 2014; Wehner et al., 2008). In brief, the site is situated about 20m above the ground level. No significant sources are found nearby. The compositions and variations of air pollutants at this site are representative of the

urban of Beijing (Guo et al., 2020; Wang et al., 2019a). The measurement was conducted from Dec 1 to Dec 31 in 2018, which was in the winter of Beijing.

#### 2.1.2  Quantification of gas-phase nitrated phenols

A chemical ionization long time-of-flight mass spectrometer (CI-LToF-MS Aerodyne Research, Inc.) equipped with a nitrate ionization source was utilized to determine the gas-phase concentration of NPs. The detailed information about the

instrumentation of CIMS can be found elsewhere (Bean and Hildebrandt Ruiz, 2016; Fang et al., 2020). Briefly, in a high purity flow of nitrogen, an X-ray source was used to ionize the reagent gas which then entered the ion-molecule region (IMR). NPs molecules reacted with these reagent ions, i.e. $NO_3^-(HNO_3)_{0-2}$, to form the product ions. Seven NPs were quantified in the present work, i.e, nitrophenol (NP, m/z 201.0153 charged with $NO_3^-$), methyl-nitrophenol (MNP, m/z 215.0310), dimethyl-nitrophenol (DMNP, m/z 229.0466), nitrocatechol (NC, m/z 217.0102), methyl-nitrocatechol (MNC,

m/z 231.0258), dinitrophenol (DNP, m/z 246.0004) and methyl-dinitrophenol (MDNP or DNOC, m/z 260.0160). The original time resolution of the concentration of NPs was 1s. The CIMS data processing was constructed by Tofware 3.0.3 (Tofwerk AG, Aerodyne Research) in Igor Pro 7.08 (WaveMetrics Inc) (Stark et al., 2015; Yatavelli et al., 2014). The chemical structures of these NPs and the results of high-resolution peak fits of reagent ions and NPs could be found in Figure S1.



### 2.1.3 Calibration of gas-phase nitrated phenols

The calibration of CIMS is challenging as a wide detection range of CIMS and unknown molecular structures of the compound detected by ToF-MS (Priestley et al., 2018). In this study, we used a Dynacalibrator® permeator (Modle 500, VICI, MetronIcs Inc.) to generate nitrophenol standard gas with high stability and accuracy. The permeation rate of the NP permeation tube (Dynacal®, VICI) is 97 ng min$^{-1}$. The standard gas-phase nitrophenol was mixed with 2 L min$^{-1}$ – 15 L min$^{-1}$ synthetic air in the permeator to create different concentrations, and then was diluted by 8 L min$^{-1}$ synthetic air. The calibration curve was made by plotting the actual gas-phase nitrophenol concentration as the function of ion signals detected (Figure S2).

### 2.1.4 Supplementary measurements

Relative humidity (RH) and temperature (T) was measured by Met one Instrument Inc. at the PKUER site. NO-NO$_2$-NO$_x$ gas analyzers (Thermo Fisher Scientific, model 42i-TLE) and UV photometric O$_3$ analyzer (Thermo Fisher Scientific model 49i) were utilized to measure the concentration of NO, NO$_2$, NO$_x$ and O$_3$. Volatile organic compounds (VOCs) were measured by an online gas chromatography-mass spectrometry/flame ionization detector (online-GC-MS/FID, Tianhong, China) (Liu et al., 2005; Shao et al., 2009). Totally 98 kinds of VOCs were measured, including alkanes, alkenes, aromatics, acetylene and oxygenated volatile organic compounds (OVOCs) which were consistent with other work (Yu et al., 2020; Yuan et al., 2013).

## 2.2 Estimation of primary sources and secondary formation of nitrated phenols

A zero-dimensional box model that functioned with the Master Chemical Mechanism (MCMv3.3.1) was utilized to simulate the secondary formation process of NPs. NPs from the oxidations of primary phenol and benzene were apportioned. The primary emission was calculated by the subtraction from the total measured concentration and then resolved by non-negative matrix factorization (NMF). The concentration weighted trajectory (CWT) analysis was also utilized to identify the source regions of the regional transport.

The data were analyzed by R 3.6.3 (R Core Team, 2020), including openair (Ropkins and Carslaw, 2012), Biobase (Huber et al., 2015), NMF (Gaujoux and Seoighe, 2010), ggplot2 (Wickham, 2016) and other necessary packages.

### 2.2.1 Estimation of secondary formation of nitrated phenols by a box model

A zero-dimensional box model functioned with the Master Chemical Mechanism (MCMv3.3.1, http://mcm.leeds.ac.uk/MCM/home) was utilized to simulate the secondary formation process of NPs. The related mechanism was presented in Figure 1. Water vapor, temperature and pressure, and the concentration of NO, NO$_2$, O$_3$, CO were used to constrain the model simulation in all scenarios. The basic model constrained the concentration of benzene, toluene and xylene measured by online GC-MS/FID. This basic model illustrated the secondary formation process of the NPs from the oxidation of aromatic hydrocarbons. However, less than 1% of the total nitrophenol (NP) concentration can be





explained (Figure S3) which was inconsistent with the estimation from NP/CO ratio in other studies (Inomata et al., 2013; Sekimoto et al., 2013), implying there are probably missing mechanisms. For instance, secondary formation of ambient NP does not only come from benzene oxidized phenol, but also originates directly from emitted phenol (so-called primary phenol). As a result, we constrained the phenol concentration rather than benzene to investigate the nitrophenol formation from primary phenol. As the concentration of primary phenol was not determined in this study, we used the ratio of

phenol/NOy (0.01-0.3 ppt/ppb) and phenol/CO (0.01-0.4 ppt/ppb) from fresh emitted vehicle exhaust (Inomata et al., 2013; Sekimoto et al., 2013). The upper value of the ratios, i.e. 0.3 ppt/ppb and 0.4 ppt/ppb were utilized, because the estimated phenol concentration in this approach was comparable to the measured concentration from other sites (Table 1). The budge analysis and the source apportionment were composed based on the constrained results of estimated phenol concentration by the ratio of phenol/CO.

**2.2.2  Source apportionment of nitrated phenols by non-negative matrix factorization (NMF)**

In this work, non-negative matrix factorization (NMF) approach was used to estimate the primary contributions of NPs. The total primary NPs was calculated by subtracting the secondary NPs from box model by the total NPs. NMF is a model that is good at dealing with multi-dimensional data, which shares the same principle with the well-known positive matrix factorization (PMF). In principle, NMF decomposes a matrix X (the concentration matrix) into two non-negative matrices W

(the source contribution matrix) and H (the source profile matrix) (Devarajan, 2008).

$$X \approx WH \tag{1}$$

Where X, W and H are n×p, n×r and r×p non-negative matrices, r is a positive integer that indicates the number of the factors. The approach of NMF is to minimize the estimation of W and H:

$$\min_{W,H \geq 0} \underbrace{[D(X, WH) + R(W, H)]}_{=F(W,H)} \tag{2}$$

Where D is the Kullback-Leibler (KL) divergence utilized in this study:

$$D: A, B \mapsto KL(A||B) = \sum_{i,j} a_{i,j} \log \frac{a_{i,j}}{b_{i,j}} - a_{i,j} + b_{i,j} \tag{3}$$

R(W, H) is an optional regularization function enforcing the constraints of W and H (Renaud and Seoighe, 2020).

NMF has been widely used in facial pattern recognition (Lee and Seung, 1999), signal and data analytics (Fu et al., 2019), and computational biology (Devarajan, 2008). Strictly speaking, PMF is a specific NMF model used in environmental

sciences (Paatero and Tapper, 1994). In recent years, NMF turns out to be a powerful technique to distinguish oxygenated organic compounds from numerous urban sources (Karl et al., 2018). Compared with PMF, NMF approach is equipped with more algorithms for matrix factorization, e.g. brunet (Brunet et al., 2004), lee (Lee and Seung, 2001), nsNMF (Pascual-Montano et al., 2006) and other methods listed on the NMF vignette (Renaud and Seoighe, 2020). Besides, the cophenetic coefficient is a fundamental way to give the optimal choice of factorization rank r while the consensus map approach avoids

overfitting. The advantage of NMF is the convincing factor choice rather than, the casual selection by PMF.



### 2.2.3 Concentration weighted trajectory (CWT) analysis

Back trajectory analysis was accomplished by the interface of Hysplit (Rolph et al., 2017; Stein et al., 2015) and R. The primary source resolved by NMF was then distinguished by the concentration weighted trajectory (CWT) approach (Seibert et al., 1994) in an attempt to identify the location of the probable source. The CWT calculated the logarithmic mean concentration of NPs for every grid as the Eq. 4. Normally, a high value of $\overline{C_{ij}}$ indicates higher concentration at the grid (i,j).

$$\ln\left(\overline{C_{ij}}\right) = \frac{1}{\sum_k^N \tau_{ijk}} \sum_k^N \ln(c_k)\tau_{ijk} \tag{4}$$

Where i and j are the indices of the grid cell (i,j), k and N are the trajectory index and the total number of trajectories, $c_k$ is the concentration of NPs when trajectory k passes by, $\tau_{ijk}$ is the resistance time of trajectory k in the cell (i,j) (Ropkins and Carslaw, 2012).

## 3  Results and discussions

### 3.1  Overview of the meteorological conditions and air pollutants

The measurement started with a heavy pollution episode from Dec 1 to Dec 2, with an average wind speed of 0.61 m s$^{-1}$, an average RH of 63%, the average concentration of PM$_{2.5}$, NO$_y$ and CO of 166 µg m$^{-3}$, 118 ppb and 1912 ppb, respectively. The average concentration of PM$_{2.5}$, NO$_y$ and CO with the heavy pollution removed was 37 µg m$^{-3}$, 49 ppb and 598 ppb, respectively. The average wind speed from Dec 3 to Dec 31 was 1.96 m s$^{-1}$ and the average RH was 20%. The heavy pollution episode accomplished with high relative humidity and slow wind speed. The time series of wind speed, RH, PM$_{2.5}$, NOy and CO during the whole sampling period may be found in Figure S4.

The concentration and composition of gas-phase NPs during the measurement were displayed in Figure 2. The average concentration of NPs (total nitrated phenols) during and without the heavy pollution episode was 1213 ± 769 ppt and 170 ± 132 ppt, while nitrophenol (NP) was the predominant species with a concentration of 662 ± 459 ppt (55%) and 97 ± 83 ppt (57%), respectively. To compare the representative NPs concentration all around the world, we evaluated the concentration in Beijing (with the episode removed) and other cities in Table 1. The concentration was converted to ng m$^{-3}$ with the aim of wide-ranging comparison. From Table 1, it was noticeable that the concentration of NPs ranged extensively from time to time with relatively higher values in winter. As for sampling sites, urban sites and those influenced by biomass burning was more likely to be polluted by NPs. Different analytical methods showed discrepancies while this may be clarified by their distinct instrumental principles. Likewise, the NPs concentration in Beijing was higher than that in rural and remote sites (Delhomme et al., 2010; Lüttke et al., 1997). Nevertheless the NPs concentration was much lower than the sites that are influenced by biomass burning (Priestley et al., 2018). The concentration of gas-phase DNP in Beijing was considerably higher than that of other sites.

Composition of NPs in Beijing during the episode and the rest period showed no significant difference, except that the proportion of DNP was 24% during the episode and 17% without the episode, respectively (Figure 2). On the contrary, a





large proportion of MNP (comparable to nitrophenol) was found in other cities (Cecinato et al., 2005; Leuenberger et al., 1988; Priestley et al., 2018). The non-negligible secondary formation of nitrophenol was a plausible explanation for this higher concentration in Beijing.

The diurnal variations of NPs were exhibited in Figure 3. Interestingly, NPs with different functional groups revealed different diurnal patterns. Nitrophenol (NP), MNP and DMNP (NPs with one -OH group and one -$NO_2$ group) demonstrated higher concentration at night and lower concentration during the day. The strong loss of gas-phase NPs due to photolysis or OH reaction during the daytime (Harrison et al., 2005; Yuan et al., 2016) might be a plausible explanation. Besides, the stable boundary layer at night might cause the accumulation of NPs as well. This indicated consistency with the studies

carried out during the UBWOS 2014 campaign (Yuan et al., 2016). Nonetheless, NC and MNC (NPs with two -OH groups and one -$NO_2$ group) displayed a small peak at about 11:00 am, which suggested a possible secondary formation process during the noon. With regards to DNP and MDNP (NPs with one -OH groups and two -$NO_2$ groups), the diurnal profiles did not vary much during the whole day except a gentle peak at about 5:00 pm and then declined at night which implied that the nighttime $NO_3$ oxidation of DNP might be a non-negligible sink.

## 200 3.2 Estimation of secondary formation and budget of nitrated phenols

In this section, gas-phase nitrophenol (NP), MNP, DMNP, DNP and MNDP were taken into account as their higher concentration and larger fraction in gas-phase. The concentration of gas-phase NC and MNC was rather low (< 4% that of nitrophenol) in this study and they were found mainly in the particle phase (Wang et al., 2019b). As a result, they were excluded from the box model results and the source apportionment.

Overall, the secondary formation accounted for 38%, 9%, 5% and 17% for ambient nitrophenol (NP), MNP, DNP and MDNP respectively. Almost 100% of DMNP could be explained by the oxidation of xylenes. The simulation results can be found in Figure S3. For nitrophenol, the simulation of the basic model and with primary phenol estimated by NOy was quite similar (both the contribution of these two model scenarios were less than 1%). When considering the primary emission of phenol by the ratio of phenol/CO (see Section 2.2.1), significant improvement of NP was found (37%). The results indicated

a sensitivity of NP production from the primarily emitted phenol so that when NPs control policies are made, it is of vital importance to control the emission of phenol rather than the classical precursor, i.e, benzene. Meanwhile, the non-linear effect of oxidation capacities and radical concentration might result in an improvement of MNP or MDNP when phenol was constrained. The model results of MDNP did not vary much as the xylene-xylenol-MDNP pathways can explain most of the secondary formation pathways of MDNP.

## 215 3.2.1 Production and loss of phenol and nitrophenol

In order to provide more insight into the secondary formation process of NPs, the production and loss analyses were conducted based on the results from the primary phenol constrained by the ratio of phenol/CO. Time series and diurnal profiles of the loss of phenol during and without the heavy pollution episode was shown in Figure 4. It was obvious that the





OH loss mainly took place during the day while NO$_3$ loss mainly happened at night. However, the fraction of these two

pathways diverged dramatically taking the episode into account. During the heavy pollution episode, 46.7% of phenol lost from the pathway which caused the production of phenoxy radical (C$_6$H$_5$O). We noticed that the C$_6$H$_5$O-NO$_2$ reaction was the only formation pathway of nitrophenol (Berndt and B öge, 2003). With the heavy pollution episode removed, the proportion of the C$_6$H$_5$O production pathway was only 5.4%. The phenol-OH reaction which produced catechol (then reacted with OH/NO$_3$, NO$_2$ to produce NC) was the predominant OH reaction (21.9%). The distinct pattern of the phenol-OH

pathway which formed C$_6$H$_5$O indicated a probable source of the nitrophenol accumulation during the heavy pollution episode. The high atmospheric reactivity and oxidation capacity in Beijing (Lu et al., 2019c; Yang et al., 2020) might be the foundation of high potential reactivity between phenol and OH radical.

The production of nitrophenol displayed two peaks at about 8:00 am and 6:00 pm while the loss remained unchanged throughout the whole day. The accumulation of nitrophenol mainly occurred in the afternoon and at night (Figure 5). The

simulation during the heavy pollution episode indicated a strong primary emission on the afternoon of Dec. 2. The production rate of nitrophenol from 12:00 am to 8:00 pm Dec 2. was lower than 10$^4$ molecular cm$^{-3}$ s$^{-1}$ with the concentration of 1357 ppt, while that during the same period on Dec 1 was higher than 2.5×10$^4$ molecular cm$^{-3}$ s$^{-1}$ with the concentration of 434 ppt. The underestimation of the box model indicated the occurrence of another source during the afternoon of Dec 2, where primary emissions might be probable.

### 235  3.2.2  Impact of secondary formation on dimethyl-nitrophenol

The box model simulation of DMNP signified the importance of the secondary formation. Production and loss of xylenol and DMNP were shown in Figure 6. The production and loss showed no distinct patterns during and without the episode. The production and loss of xylenol displayed peaks at 12:00 am and 1:00 pm respectively. The vicarious peaks lead to the accumulation of xylenol at noon. The reactions with OH and NO$_3$ radicals accounted for 42.6% and 42.5% of the loss of

xylenol. The OH reaction pathway was the predominant loss of xylenol during the daytime and resulted in the formation of DMNP. As for DMNP, the production increased rapidly from the xylenol-NO$_2$ reaction during the daytime and decreased from noon. The loss of DMNP increased during the afternoon and started to decrease after 6:00 pm. DMNP mainly originated from the secondary formation process and its accumulation mainly took place in the afternoon while nitrophenol mainly occurred at night which hailed largely from primary emission.

### 245  3.3  Source apportionment of primarily emitted nitrated phenols and the impact of regional transport

NMF approach equipped with Brunet, KL, offset, lee, nsNMF and snmf/l algorithms were used to investigate the sources of primary emitted NPs. These different algorithms were used to choose a better calculation method for the source apportionment. The consensus maps of the simulation were displayed in Figure S5. The KL approach was chosen as its well-estimated pattern. Besides, 3 to 7 factors were tested by NMF so as to get an optimal one. The NMF rank survey was shown

in Figure S6, by which four factors were chosen.


The mixture coefficients of KL algorithm with the factor of 4 was displayed in Figure 7. $SO_2$ was the tracer of factor 1 while aromatics (mainly toluene, xylene and ethylbenzene) were markers of factor 2. Chloromethane was the tracer of factor 3 while acetylene, trans-2-butene, 1,3-butadiene were markers of factor 4. The diurnal patterns of the resolved sources were displayed in Figure S7. Combined with results from the markers and the diurnal profiles of the sources, we identified these

factors as coal combustion, industry, biomass burning and vehicle exhaust. As 30.4% of DNP and 9.2% of MDNP came from factor 2, the pesticide industry was the most probable contributor.

The source contribution of NPs combining primary emission and secondary formation was displayed in Figure 8. 58% of the total NP concentration originated from biomass burning while 2.4% derived from vehicle exhaust. 76.2%, 11.8% and 1.9% of the total MNP concentration came from biomass burning, coal combustion and vehicle exhaust, respectively. As for DNP

and MDNP, despite that 64.9% and 45.8% of them were derived from biomass burning, 30.4% and 9.2% of DNP and MDNP concentration resulted from industrial emissions. This suggested that the pesticide industry was still an important source of dinitrophenols.

When coal combustion and biomass burning were regarded as combustion sources, the four-factor results of NMF, as well as the same species in PMF, were comparable (Figure S8). Combustion source account for 61.5%, 91%, 10.2% and 38% of NP,

MNP, DNP and MNDP concentration, respectively. Meanwhile, 80.1% and 45.3% of DNP and MDNP concentration were derived from industry.

Overall, the contribution of primary emission was more important than secondary formation during the measurement. Among all sources, combustion was the predominant one (>50% of total NPs concentration), which was consistent with other studies focused on the sources of particulate matter (PM) in the winter of Beijing (Fan et al., 2018; Lyu et al., 2019; Xu

et al., 2018). This result was different from the study carried out during the UBWOS 2014 (Yuan et al., 2016) in which less than 2% of NP concentration came from combustion sources. UBWOS 2014 was carried out at an oil and gas production site abundant of the precursors of NPs, i.e. VOCs (such as benzene and toluene) and $NO_x$. Therefore, the secondary process was indeed the predominant one in UBWOS 2014. By contrast, the PKUERS site was far away from industrial zones and combustion sources and was more likely to be influenced by primary emission which came from regional transport nearby.

In this study, the concentration weighted trajectory (CWT) was used to identify the probable source of these primary emissions. Considering the different pollution patterns of the sampling period as well as the amount of data for interpolation in CWT, we divided the sampling period into four sub-periods, i.e. Dec. 1-10, Dec. 10-15, Dec. 15-20, and Dec. 20-30. CWT analysis was conducted for each period, and the results were displayed in Figure 9. Strong regional transport was observed during the first period. The biomass burning and industry sources mainly originated from provinces surrounding the Yellow

and Bohai Seas (especially Tianjin City and Shandong Province). Cities located in this area had a long history of pesticide production and use and have been reported to reveal a relatively high residual concentration of pesticide (Li et al., 2018). As for the vehicle exhaust source, local emissions were predominant. The coal combustion source mainly came from Inner Mongolia, which was a coal abundant area across China (Lv et al., 2020). The CWT analysis proved the accuracy of NMF source apportionment and demonstrated the importance of regional transport when NPs control strategies were made.


The estimation of secondary formation and primary emission of NPs in this study faced uncertainties as below. The simulation of NPs in this study was restricted by the estimation of phenol, the mechanisms of MCM, as well as the simulation of $NO_3$ radical in winter. The box model results of NPs were not identical to secondary formation and the estimation of primary emissions by subtracting the NPs from the box model results by the total concentration remained uncertain. Further studies should be focused on the online phenol measurement, and the improvement of the secondary

formation mechanisms.

## 4  Conclusions

Gas-phase nitrated phenols (NPs) were measured by using a CI-LToF-CIMS in the winter of Beijing. The NPs concentrations in winter of Beijing are high with the total concentration of $1158 \pm 892$ ng m$^{-3}$, which are higher than those in most of the rural and remote sites all around the world. Nitrophenol was the predominant compound with an average

concentration of $606.3 \pm 511.1$ ng m$^{-3}$. Strong diurnal patterns were observed and NPs with different functional groups varied significantly. Nitrophenol displayed higher concentration at night and lower concentration during the day. A box model was utilized to simulate the secondary formation of NPs. 38%, 9%, 5%, 17% and almost 100% of the ambient nitrophenol (NP), MNP, DNP, MDNP and DMNP could be explained by the oxidation of aromatic precursors. The oxidation of primary phenol estimated by the ratio of phenol/CO from fresh vehicle exhaust accounted for 37% of the total nitrophenol,

while <1% might be explained by the oxidation of benzene. The latter pathway was widely used in models and might lead to great uncertainties as the primarily emitted phenol was not considered. Meanwhile, control strategies focus on the primarily emitted phenol might be more important than benzene when NPs control schemes were made. Besides, during the heavy pollution episode, 46.7% of phenol lost from the pathway of OH reaction to form phenoxy radical ($C_6H_5O$). The phenoxy radical consequently reacted with $NO_2$, and produced nitrophenol. During the non-polluted period, the reaction that produced

catechol (then produced NC) was the predominant phenol loss pathway by OH reaction (21.9%). This stressed that the phenol-$C_6H_5O$ pathway might play a role in the nitrophenol accumulation during the heavy pollution episode. Primary source apportionment was conducted by NMF (KL algorithm) model with a factor of 4. Combustion was the predominant source of primary NPs, yet 30.4% of DNP and 9.2% of MDNP were from non-combustion sources, i.e. industry. The CWT analysis indicated a probable regional transport of combustion and the industry source from provinces that surround the

Yellow and Bohai Seas.

In conclusion, nitrated phenols in winter of Beijing were mainly influenced by primary emissions and regional transport, yet secondary formation cannot be neglected. This study firstly stressed that primary emitted phenol rather than benzene oxidation was crucial in the rapid accumulation of NPs during the heavy pollution episode in Beijing. This result provides more insight into NPs pollution control strategies. Both primary sources and secondary formation in either local or regional

scale should be considered when making NPs control policies in North China.

**The Supplement related to this article is available online at DOI:**





***Acknowledgments.*** The work was funded by the National Key Research and Development Program of China (2016YFC0202000, 2017YFC0213000), National Natural Science Foundation of China (No. 41977179, 91844301, 51636003), Beijing Municipal Science and Technology Commission (Z201100008220011), and Natural Science Foundation 320 of Beijing (No. 8192022).

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



**Table 1. The concentration of phenol and nitrated phenols (NPs) in different sampling sites and their site categories, sampling time and analytical methods (ng m$^{-3}$).**

| Sampling site | Site category | Sampling time | Method | phenol | NP | DNP | MNP | DMNP | NC | MDNP | MNC | Reference |
|---|---|---|---|---|---|---|---|---|---|---|---|---|
| Strasbourg area, Francev | urban and rural sites | annual mean | GC-MS | 0.4-58.7 | 0.01-2.2 | 5.6 | 2.6 | | | 0.1-0.3 [a] | | (Delhomme et al., 2010) |
| Rome, Italy | downtown | winter-spring | GC-MS | | 14.3 | | 13.9 | 2.0 (1.0) [b] | | | | (Cecinato et al., 2005) |
| Great Dun Fell, England | remote site | spring | GC-MS | 14-70 | 2-41 [c] | 0.1-8.5 | | | | 0.2-6.6 | | (Lüttke et al., 1997) |
| Beijing, China | regional site | spring | LC-MS | | *143-566* [d] | | *7.1-62* [e] | | *0.06-0.79* [f] | | *0.017* [g] | (Wang et al., 2019b) |
| Milan, Italy | polluted urban site | summer | HPLC | 400 | 300 | | | | | | | (Belloli et al., 1999) |
| northern Sweden | dairy farms | autumn-winter | TD-GC | 3000-50000 | | | | | | | | (Sunesson et al., 2001) |
| Manchester, UK | with Bonfire Plume Removed | autumn-winter | ToF-CIMS | | 780 | | 630 | | | | | (Priestley et al., 2018) |
| Ottawa, Canada | selected dwellings sites | winter | TD-GC-MS | 10-1410 | | | | | | | | (Zhu et al., 2005) |
| Santa Catarina, Brazil | near a coal-fired power station | winter | GC-FID | 980-1600 | | | | | | | | (Moreira Dos Santos et al., 2004) |
| Switzerland | urban site | winter | GC-MS | 40 | 350 [h] | | 250 [i] | | | 50 [j] | | (Leuenberger et al., 1988) |
| Manchester, UK | measured during the | winter | ToF-CIMS | | 3700 | | 3600 | | | | | (Priestley et al., |




| | | | | | | | | | | | | |
|---|---|---|---|---|---|---|---|---|---|---|---|---|
| | bonfire night | | | | | | | | | | | 2018) |
| Detling, United Kingdom | rural site | winter | MOVI-HR ToF-CIMS | | 0.02 | 3 | 5 | | | 2.5 | | 8.2 | (Mohr et al., 2013) |
| Beijing, China (this study) | urban site | winter | ToF-CIMS | *63*[k] *1013*[l] | 606.3 (511.1) | 243.5 (339.6) | 203.5 (156.6) | 46.2 (32.6) | 22.1 (12.4) | 26.0 (25.8) | 10.4 (6.3) | |

The estimated concentration was displayed in the *italic* script while standard variation was displayed in brackets. Nitrated phenols investigated in this study referred to nitrophenol (NP), dinitrophenol (DNP), methyl-nitrophenol (MNP), dimethyl-nitrophenol (DMNP), nitrocatechol (NC), methyl-dinitrophenol (MDNP) and methyl-nitrocatechol (MNC).

[a] gas+particle phase; [b] 2,6-Dimethyl-4-nitrophenol; [c] 2/4-Nitrophenol; [d] 4NP, estimated; [e] 2M4NP+3M4NP, estimated; [f] 4NC, estimated; [g] 3M6NC+3M5NC+4M5NC, estimated; [h] 2-Nitrophenol; [i] 3M2NP+4M2NP; [j] 2,4-Dinitro-6-methyl phenol;

[k] estimated by 0.3NOy; [l] estimated by 0.4CO





**Figures Caption**

**Figure 1.** Mechanism related to the secondary formation of the nitrated phenols (NPs) in MCM 3.3.1 applied in this study.
Different model scenarios differed in the constraints of the precursors. The basic model constrained the concentration of benzene by measurement from online GC-MS/FID. The other model scenarios constrained primary phenol concentration rather than benzene estimated by the ratio of phenol/NOy or phenol/CO from fresh vehicle exhaust.

**Figure 2.** Time series (local time) and compositions of nitrated phenols (NPs) during the heavy pollution episode (Dec 1 and Dec 2) and with the heavy pollution episode removed (Dec 3 to Dec 31). NPs in the legend referred to nitrophenol (NP),
dinitrophenol (DNP), methyl-nitrophenol (MNP), dimethyl-nitrophenol (DMNP), nitrocatechol (NC), methyl-dinitrophenol (MDNP) and methyl-nitrocatechol (MNC).

**Figure 3.** Diurnal profiles of nitrated phenols (NPs) with 95% confidence interval in the mean. The concentration of NPs was normalized by their mean values. (a) Diurnal profiles of nitrophenol (NP), methyl-nitrophenol (MNP) and dimethyl-nitrophenol (DMNP). These are NPs with one -OH group and one -NO$_2$ group, (b) Diurnal profiles of nitrocatechol (NC) and
methyl-nitrocatechol (MNC). These are NPs with two -OH groups and one -NO$_2$ group), (c) Diurnal profiles of dinitrophenol (DNP) and methyl-dinitrophenol (MDNP). These are NPs with one -OH groups and two -NO$_2$ groups.

**Figure 4.** Time series and the loss rate of phenol during the heavy pollution episode (a) and diurnal profile of the loss of phenol with the heavy pollution removed (b).

**Figure 5.** Time series of production and loss of nitrophenol (NP) during the heavy pollution episode (a) and diurnal profiles
of production and loss of NP with the heavy pollution removed (b).

**Figure 6.** Production and loss of xylenol (a) and DMNP (b) during the sampling period.

**Figure 7.** Mixture coefficients of the Kullback-Leibler (KL) algorithm with the factor number of four by non-negative matrix factorization (NMF). Factor 1: coal combustion; Factor 2: industry (pesticide); Factor 3: biomass burning; Factor 4: vehicle exhaust. *Basis* and *consensus* in the legend were the model runs in which the latter one was consensus and the results
were displayed in the heatmap.

**Figure 8.** Contribution of primary emission (in blue borderline) and second formation (in red borderline) of nitrated phenols. Primary emission was classified as biomass burning, coal combustion industry and vehicle exhaust which were resolved by non-negative matrix factorization (NMF). NPs in the legend referred to dinitrophenol (DNP), methyl-dinitrophenol (MDNP), methyl-nitrophenol (MNP), and nitrophenol (NP). Secondary formation of nitrophenol was categorized as benzene oxidation
(<1%) and the oxidation of primarily emitted phenol (phenol oxidation, 37%). It was noticeable that nitrophenol derived from the primary phenol oxidation was much more important than the pathway from the traditional benzene oxidation in winter of Beijing.

**Figure 9.** Concentration weighted trajectory (CWT) analysis of the sources resolved by non-negative matrix factorization (NMF), i.e, coal combustion (a), biomass burning (b) industry (c) and vehicle exhaust (d).








**Figure 1.** Mechanism related to the secondary formation of the nitrated phenols (NPs) in MCM 3.3.1 applied in this study. Different model scenarios differed in the constraints of the precursors. The basic model constrained the concentration of benzene by measurement from online GC-MS/FID. The other model scenarios constrained primary phenol concentration rather than benzene estimated by the ratio of phenol/NOy or phenol/CO from fresh vehicle exhaust.





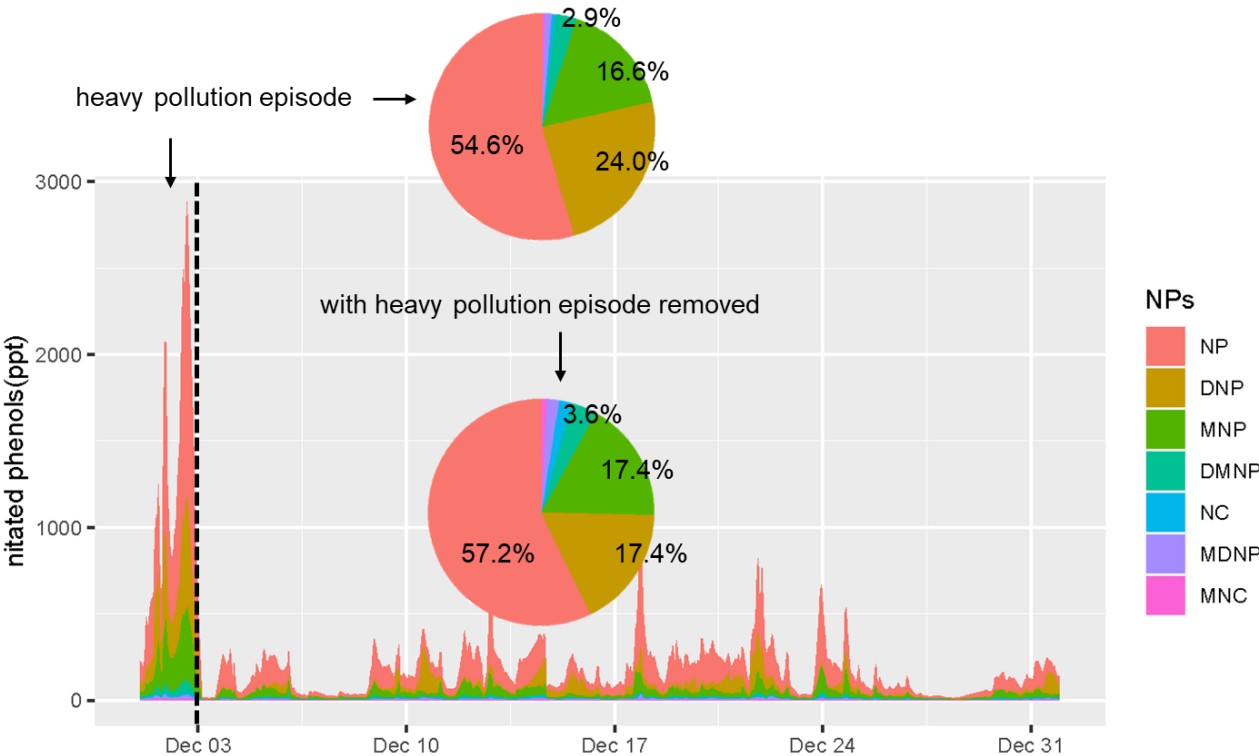

**Figure 2.** Time series (local time) and compositions of nitrated phenols (NPs) during the heavy pollution episode (Dec 1 and
Dec 2) and with the heavy pollution episode removed (Dec 3 to Dec 31). NPs in the legend referred to nitrophenol (NP),
dinitrophenol (DNP), methyl-nitrophenol (MNP), dimethyl-nitrophenol (DMNP), nitrocatechol (NC), methyl-dinitrophenol
(MDNP) and methyl-nitrocatechol (MNC).

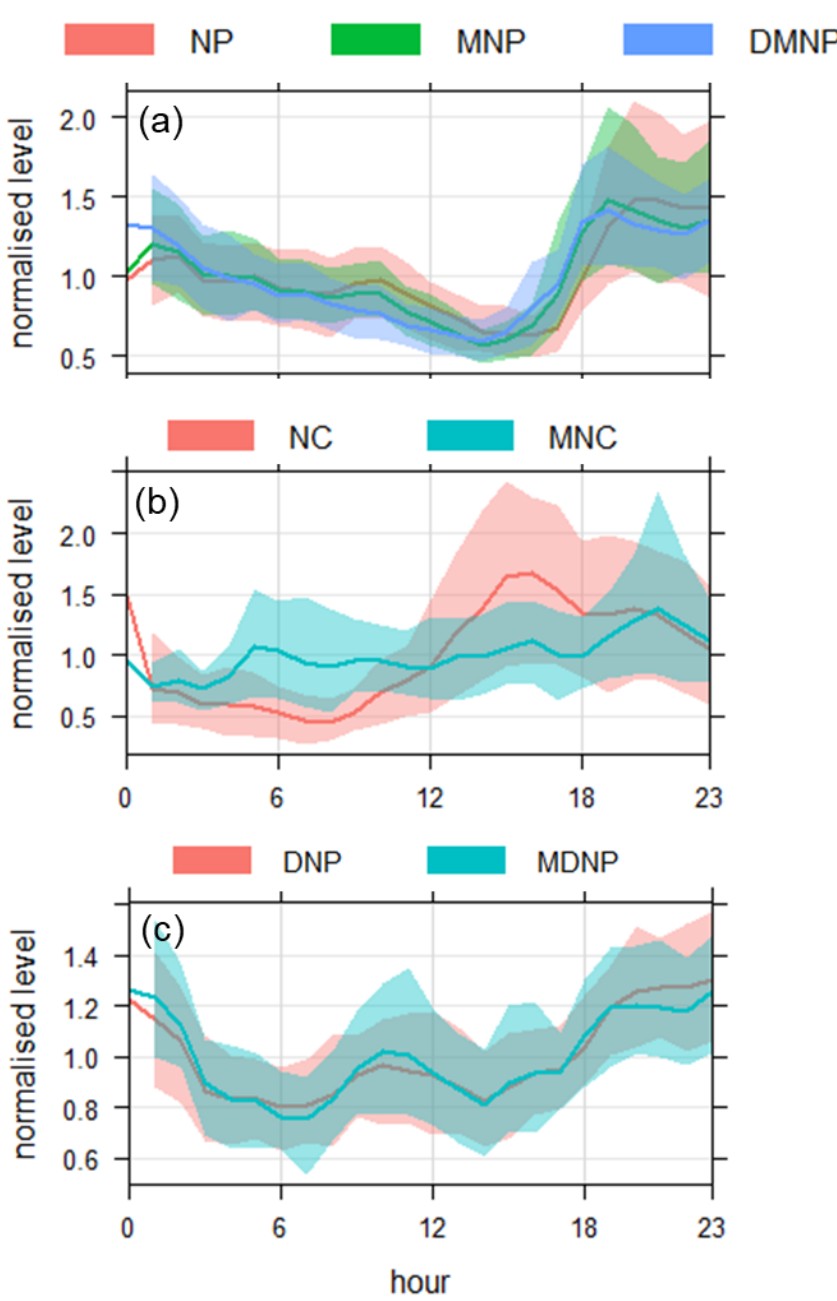

**Figure 3.** Diurnal profiles of nitrated phenols (NPs) with 95% confidence interval in the mean. The concentration of NPs was normalized by their mean values. (a) Diurnal profiles of nitrophenol (NP), methyl-nitrophenol (MNP) and dimethyl-nitrophenol (DMNP). These are NPs with one -OH group and one $-NO_2$ group, (b) Diurnal profiles of nitrocatechol (NC) and methyl-nitrocatechol (MNC). These are NPs with two -OH groups and one $-NO_2$ group), (c) Diurnal profiles of dinitrophenol (DNP) and methyl-dinitrophenol (MDNP). These are NPs with one -OH groups and two $-NO_2$ groups.



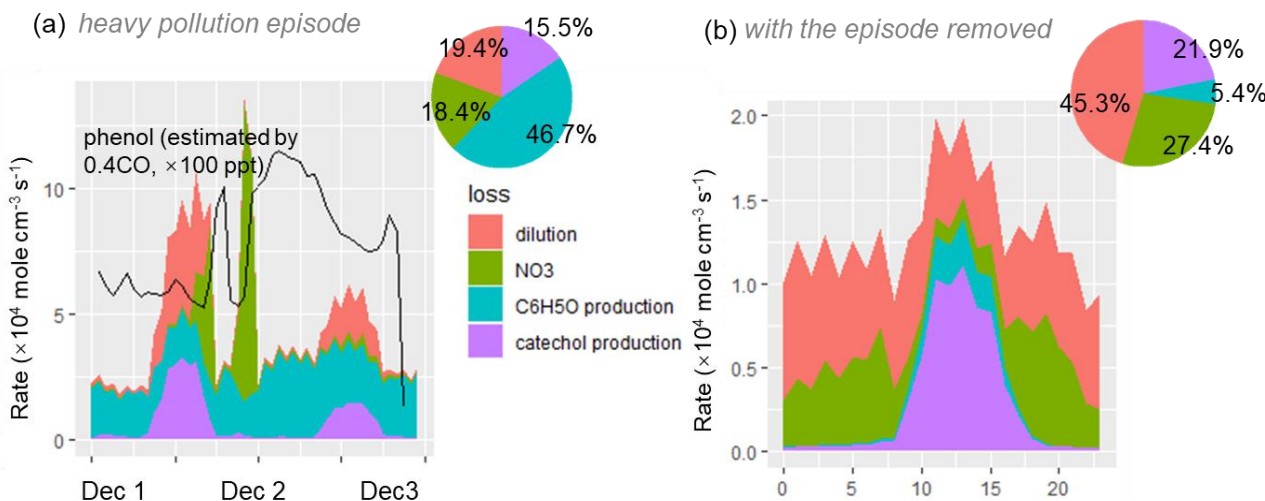

**Figure 4.** Time series and the loss rate of phenol during the heavy pollution episode (a) and diurnal profile of the loss of phenol with the heavy pollution removed (b).

575





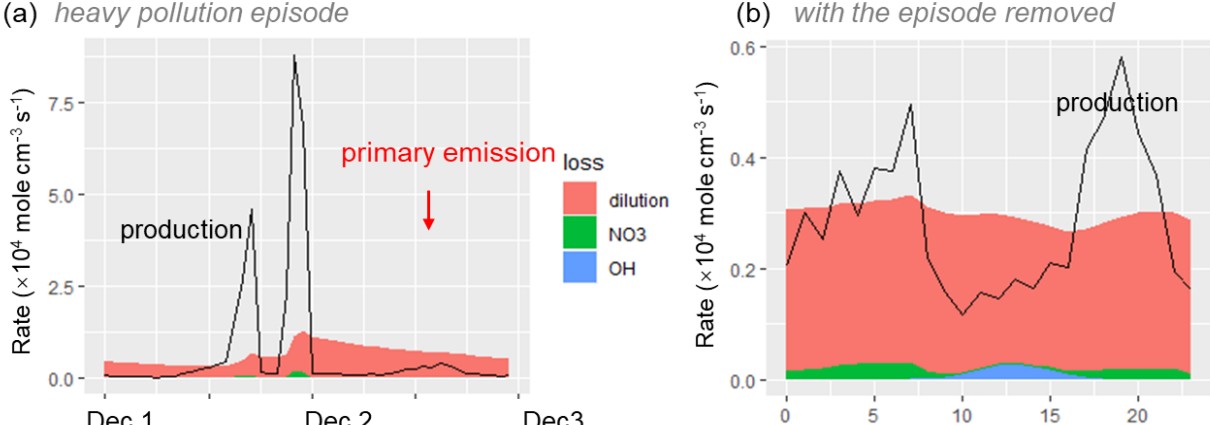

**Figure 5.** Time series of production and loss of nitrophenol (NP) during the heavy pollution episode (a) and diurnal profiles of production and loss of NP with the heavy pollution removed (b).

580





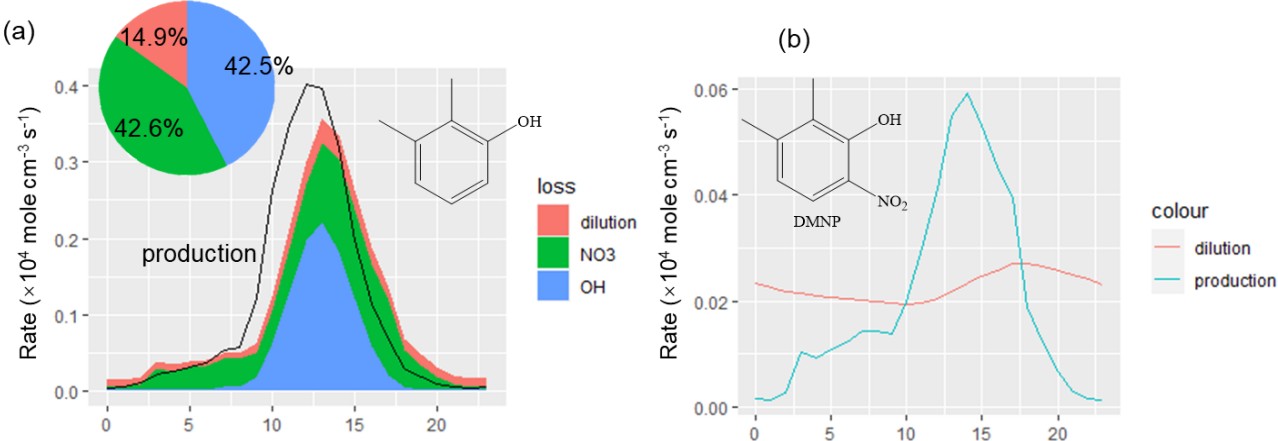

**Figure 6.** Production and loss of xylenol (a) and DMNP (b) during the sampling period.

585





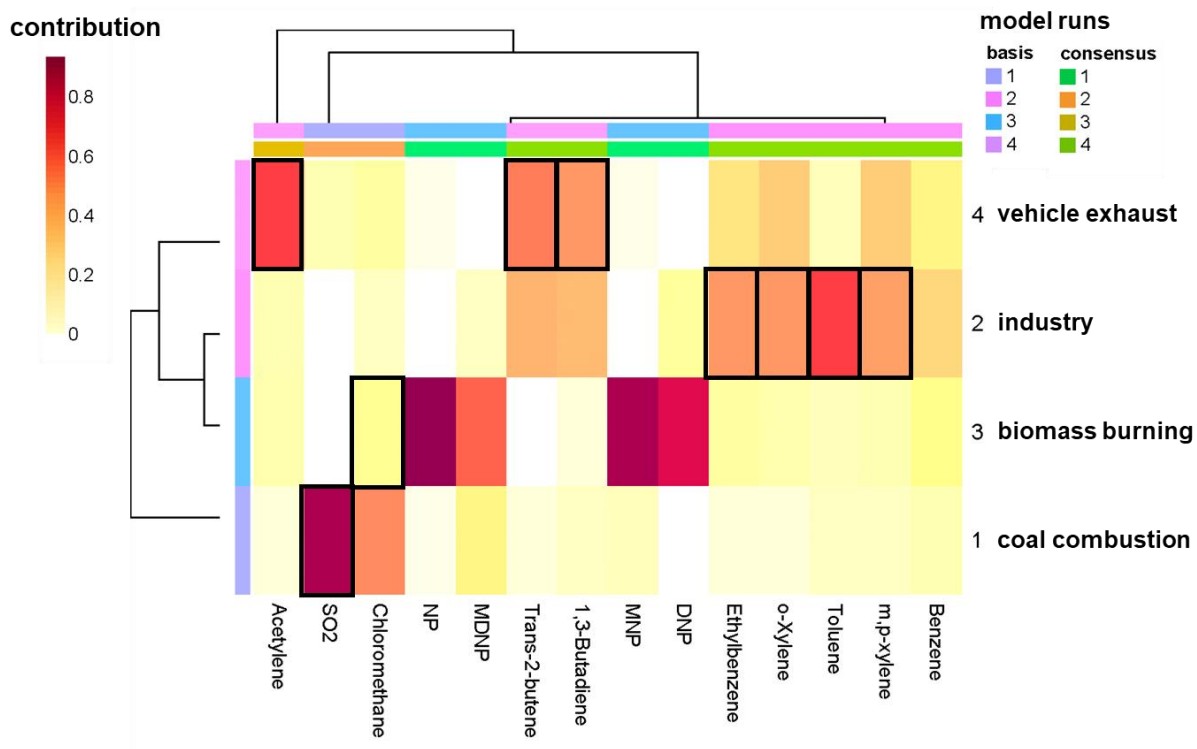

**Figure 7.** Mixture coefficients of the Kullback-Leibler (KL) algorithm with the factor number of four by non-negative matrix factorization (NMF). Factor 1: coal combustion; Factor 2: industry; Factor 3: biomass burning; Factor 4: vehicle exhaust. *Basis* and *consensus* in the legend were the model runs in which the latter one was consensus and the results were displayed in the heatmap.





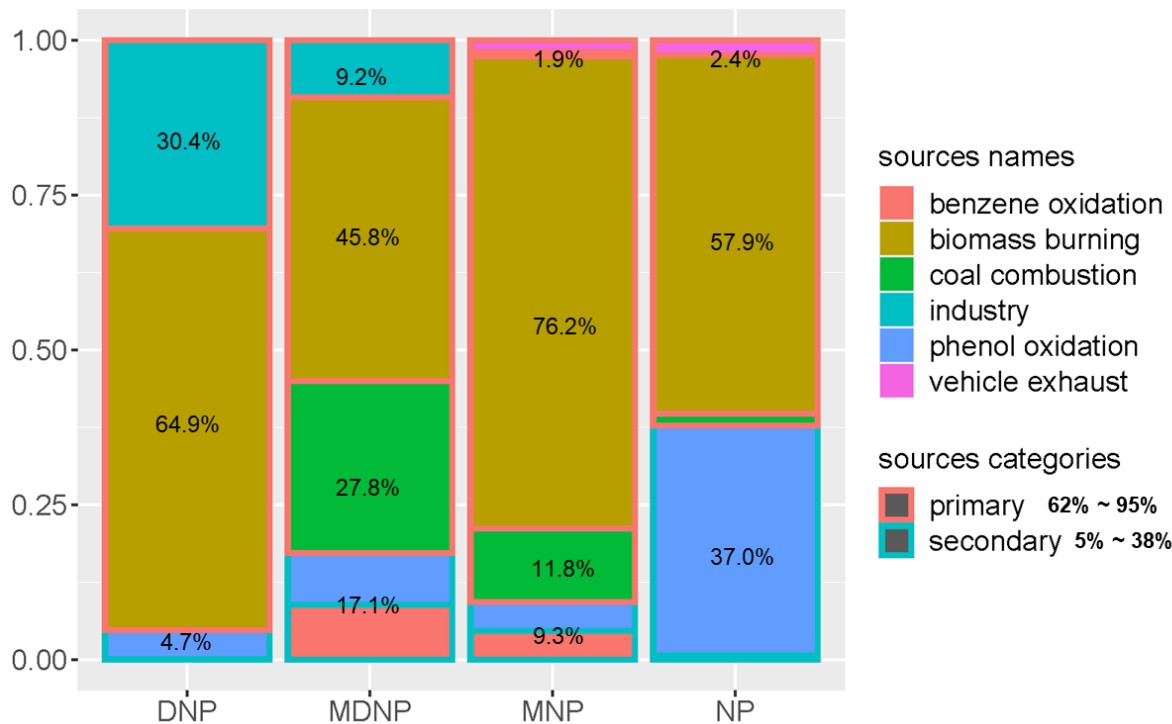

**Figure 8.** Contribution of primary emission (in blue borderline) and second formation (in red borderline) of nitrated phenols.
Primary emission was classified as biomass burning, coal combustion industry and vehicle exhaust which were resolved by
non-negative matrix factorization (NMF). NPs in the legend referred to dinitrophenol (DNP), methyl-dinitrophenol (MDNP),
methyl-nitrophenol (MNP), and nitrophenol (NP). Secondary formation of nitrophenol was categorized as benzene oxidation
(<1%) and the oxidation of primarily emitted phenol (phenol oxidation, 37%). It was noticeable that nitrophenol derived
from the primary phenol oxidation was much more important than the pathway from the traditional benzene oxidation in
winter of Beijing.

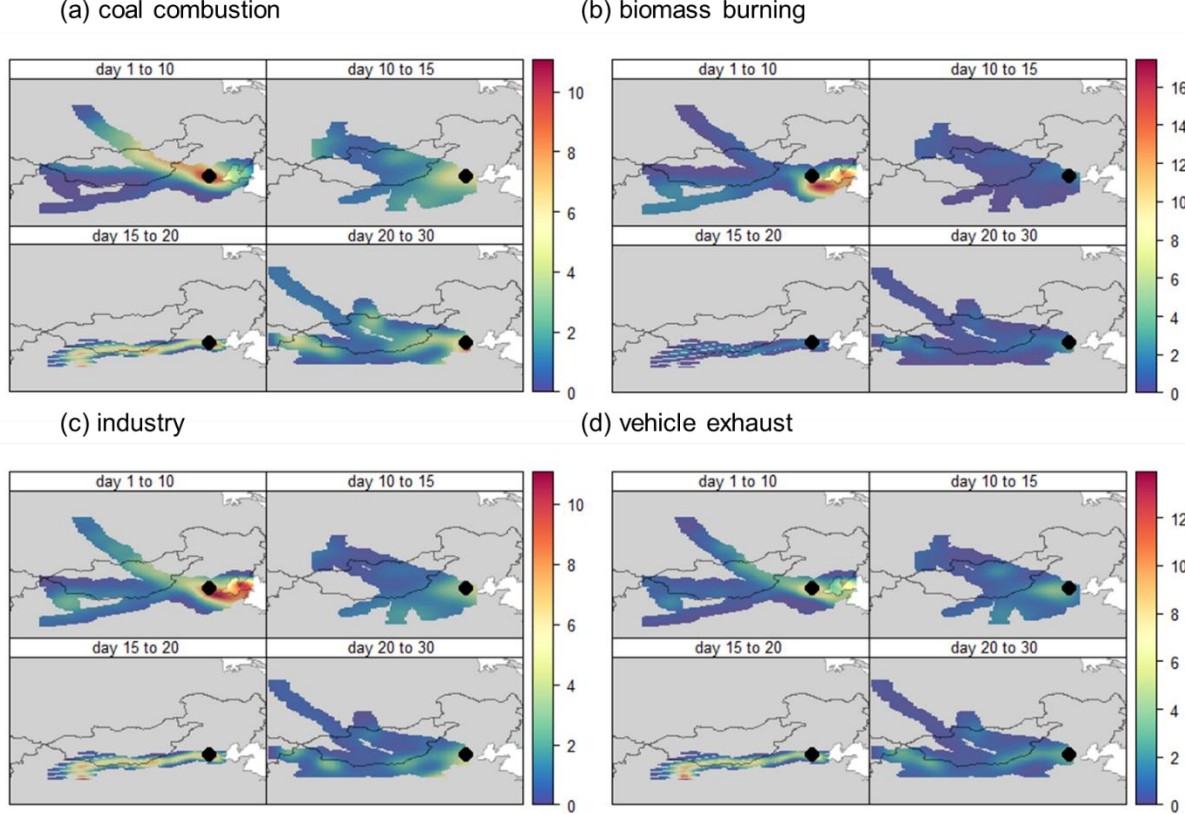

**Figure 9.** Concentration weighted trajectory (CWT) analysis of the sources resolved by non-negative matrix factorization
(NMF), i.e, coal combustion (a), biomass burning (b) industry (c) and vehicle exhaust (d).