# Peer review of "Measurement Report: Online Measurement of Gas-Phase Nitrated Phenols Utilizing CI-LToF-MS: Primary Sources and Secondary Formation"

_Atmospheric Chemistry and Physics, 2020_

## Author Comment (AC1)

We thank the reviewers for their careful review of our manuscript. The comments and suggestions greatly improve our manuscript. Following is our point to point responses to the comments:

**Response to referee #1:**

This paper describes an interesting analysis of the sources and formation of nitrated phenolic compounds in a Mega city. The material presented is original and the topics are well chosen. The paper contains some model-based data analysis parts and a section about source apportionment by

NMF. While the latter is quite well done the former has some room for improvements. Actually, the manuscript is in these parts difficult to follow. I think there are two reasons for that. There may be some weaknesses in the language (however, I am not a native speaker myself) and possibly some errors and un-preciseness in relation of the figures and their description in the text. Both together made it difficult to really judge the quality of the scientific content.

Still, in my opinion, the manuscript has valuable information and potentially good science in it. I

suggest, to consider the manuscript for publication in ACP after some major revisions and formal improvements along the comments below.

We thank the referee for the careful review and valuable suggestions. We have asked a native speaker to help us to go through the manuscript.

**Formal:**

I suggest the authors a) check use of present time / past tense; b) check use singular / plural for predicates / verbs; c) check use of single words and notations (in a thesaurus), if they really express what they wanted to say. In addition, it seems that names are mixed up, which makes it difficult and time consuming to understand the results. I indicated some examples below, but not all.

We thank the reviewer for the comments. We go through the text to check the expressions and grammar. In addition, We have asked a native speaker to help us to edit the manuscript.

**Major comments:**

line 126 – 131: The authors notate missing "mechanism" for NP formation. But NP from phenol oxidation is in their chemical mechanism, so I would call it missing "sources of phenol". And that is actually, how they treat the problem, by testing phenol sources with two different source strength.

We agree with the reviewer that "source of phenol" is more appropriate here.

The manuscript has been revised as follows (line 128-130): "However, less than 1% of the total nitrophenol (NP) concentration could be explained (Figure S3) which was inconsistent with the estimation from NP/CO ratio in other studies, implying there are probably missing sources of phenol."

At the same time, I am wondering what is to be learned from the use of the two suggested emission factors of phenol. The emission ratios phenol/NOY and phenol/CO look similar, but de facto they lead to an order of magnitude different phenol concentrations, because of the different concentrations of NOY and CO. Do the authors want to say that linking phenol to CO leads to more realistic phenol concentrations?   However, the use of the CO related phenol source leads indeed to about the right level NP concentrations, but the model time series does not really match the time series of observations.

Thank you for your comment. The concentrations of NOy and CO play an important role in phenol estimation. However, the atmospheric behaviors of NOy and CO are more important. VOCs to CO

ratio is widely used to quantify anthropogenic emissions because the atmospheric CO is inert (Li et al., 2019). In this study, we linked the phenol to CO in order to identify a more reliable estimation for phenol concentrations. We agree that the time series of the model estimation by phenol/CO ratio showed discrepancies in some days when nitrophenol concentration is low. Nevertheless, the trend and concentration level of NP and DNP (Figure S3) showed good agreement in polluted days when using phenol to CO. However, when using phenol/NOy ratio, there shows large discrepancies. NMF

and CWT analysis revealed the importance of primary emissions during the heavy pollution period and estimation from freshly emitted CO could be more reasonable. We add some detailed explanation in the main text to show why we use phenol to CO ratio to estimate phenol concentration as follows (line 133 - 139):

As the concentration of primary phenol was not determined in this study, we used the ratio of phenol/NOy (0.3 ppt/ppb) and phenol/CO (0.4 ppt/ppb) from fresh emitted vehicle exhaust (Inomata et al., 2013; Sekimoto et al., 2013). Atmospheric CO is inert so that VOCs to CO ratio is widely used to quantify anthropogenic emissions (Li et al., 2019).   The estimation of phenol from phenol/CO ratio showed good agreement in trend and concentration level (Figure S3). The estimated phenol concentration in this approach was comparable to the measured concentration from other sites (Table 1). As a result, the budget analysis and the source apportionment were composed based on the constrained results of estimated phenol concentration by the ratio of phenol/CO.

In addition, unfortunately, the most important last lines in Table 1 are messed up. What is the meaning of the number in brackets? I would also suggest, to replace the different references in the last column of Table 1 by numbers or symbols and list them in the captions under the table.

Thank you for your comment. The meaning of the number in brackets are the standard deviations of the concentrations in Table 1 which was demonstrated in the table caption. We revised the table caption to make it more clear to readers. The revision is as follows in line 513, "The estimated concentrations were displayed in the italic script. Standard variations were displayed in brackets." The references are replaced by numbers in the manuscript (line 545 - 547).

**Table 1. The concentration of phenol and nitrated phenols (NPs) in different sampling sites and their site categories, sampling time and analytical methods (ng m$^{-3}$).**

| Sampling site | Site category | Sampling time | Method | phenol | NP | DNP | MNP | DMNP | NC | MDNP | MNC | References |
|---|---|---|---|---|---|---|---|---|---|---|---|---|
| Strasbourg area, France | urban and rural sites | annual mean | GC-MS | 0.4-58.7 | 0.01-2.2 | 5.6 | 2.6 | | | 0.1-0.3 [a] | | 1 |
| Rome, Italy | downtown | winter-spring | GC-MS | | 14.3 | | 13.9 | 2.0 (1.0) [b] | | | | 2 |
| Great Dun | remote site | spring | GC-MS | 14-70 | 2-41 [c] | 0.1-8.5 | | | | 0.2-6. | | 3 |

| Location | Site | Season | Method | | | | | | | | | Ref |
|---|---|---|---|---|---|---|---|---|---|---|---|---|
| Fell, England | | | S | | | | | | | 6 | | |
| Beijing, China | regional site | spring | LC-MS | | *143-566* [d] | | *7.1-62* [e] | | *0.06-0.79* [f] | | *0.017* [g] | 4 |
| Milan, Italy | polluted urban site | summer | HPLC | 400 | 300 | | | | | | | 5 |
| northern Sweden | dairy farms | autumn-winter | TD-GC | 3000-50000 | | | | | | | | 6 |
| Manchester, UK | with Bonfire Plume Removed | autumn-winter | ToF-CIMS | | 780 | | 630 | | | | | 7 |
| Ottawa, Canada | selected dwellings sites | winter | TD-GC-MS | 10-1410 | | | | | | | | 8 |
| Santa Catarina, Brazil | near a coal-fired power station | winter | GC-FID | 980-1600 | | | | | | | | 9 |
| Switzerland | urban site | winter | GC-MS | 40 | 350 [h] | | 250 [i] | | | 50 [j] | | 10 |
| Manchester, UK | measured during the bonfire night | winter | ToF-CIMS | | 3700 | | 3600 | | | | | 7 |
| Detling, United Kingdom | rural site | winter | MOVI-HR-ToF-CIMS | | 0.02 | 3 | 5 | | 2.5 | | 8.2 | 11 |
| Beijing, China (this study) | urban site | winter | ToF-CIMS | *63* [k] / *1013* [l] | 606.3 (511.1) | 243.5 (339.6) | 203.5 (156.6) | 46.2 (32.6) | 22.1 (12.4) | 26.0 (25.8) | 10.4 (6.3) | |

The estimated concentrations were displayed in the *italic* script. Standard variations were displayed in brackets. Nitrated phenols investigated in this study referred to nitrophenol (NP), dinitrophenol (DNP), methyl-nitrophenol (MNP), dimethyl-nitrophenol (DMNP), nitrocatechol (NC), methyl-dinitrophenol (MDNP) and methyl-nitrocatechol (MNC).

**Symbols:** [a] gas+particle phase; [b] 2,6-Dimethyl-4-nitrophenol; [c] 2/4-Nitrophenol; [d] 4NP, estimated; [e]

2M4NP+3M4NP, estimated; [f] 4NC, estimated; [g] 3M6NC+3M5NC+4M5NC, estimated; [h]

2-Nitrophenol; [i] 3M2NP+4M2NP; [j] 2,4-Dinitro-6-methyl phenol; [k] estimated by 0.3NOy; [l] estimated by 0.4CO

**References:** [1] (Delhomme et al., 2010); [2] (Cecinato et al., 2005); [3] (Lüttke et al., 1997); [4] (Wang et al., 2019); [5] (Belloli et al., 1999); [6] (Sunesson et al., 2001); [7] (Priestley et al., 2018); [8] (Zhu et al.,

2005); [9] (Moreira Dos Santos et al., 2004); [10] (Leuenberger et al., 1988); [11] (Mohr et al., 2013).

line 195-199: I am sorry, I am not able to recognize the features described in the manuscript for the

Figure b and c. E.g. NC and MNC have a different diurnal cycle but are treated together. I can also not identify gentle peaks at 5 pm.   To me it looks as if either the descriptions do not express what is intended to say or the explanations and plots maybe mixed up.

It would be also helpful if the time notations in the manuscripts and at the axis of the Figure would be the same and to have minor ticks at the time axis or a grid in the diagram.

Because of all this I cannot really judge conclusions drawn from diurnal cycles.

Thank you for your comment. The revised diurnal profiles of nitrated phenols were displayed in

Figure 3 in the manuscript, with clear axes, ticks, and grids. The different diurnal cycles of DNP and

MDNP are also separated. The revised sentences are as follows,

"Nonetheless, NC and MNC (NPs with two -OH groups and one -NO$_2$ group) displayed a small peak at about 10:00 am, and revealed high concentrations at night. DNP and MDNP (NPs with one -OH

groups and two -NO$_2$ groups) displayed distinct patterns from either NP or NC. DNP accumulated during the afternoon and began to decline after 5:00 p.m., suggesting that NO$_3$ oxidation of DNP

might be a non-negligible sink. The diurnal profile of MDNP did not vary much during the whole day with a slight increase at night" (line 200 - 204).

[Figure]

**Figure 3.** Diurnal profiles of nitrated phenols (NPs) with 95% confidence interval in error bars. The concentration of NPs was normalized by their mean values. (a) Diurnal profiles of nitrophenol (NP), methyl-nitrophenol (MNP) and dimethyl-nitrophenol (DMNP). These are NPs with one -OH group and one -NO$_2$ group. (b) Diurnal profiles of nitrocatechol (NC) and methyl-nitrocatechol (MNC). These are NPs with two -OH groups and one -NO$_2$ group). Diurnal profiles of (c) dinitrophenol (DNP) and (d) methyl-dinitrophenol (MDNP). These are NPs with one -OH groups and two -NO$_2$ groups.

line 205 – 214: Again, I have difficulties to follow the text along the Figure S3. If DMNP is explained by the xylene emissions the red symbols should indicate that, because this should be covered by the base case, right? I don't see them. On the other hand, MDNP is according Figure 1 a product of toluene, not of xylene, as I think, is claimed in line 213. In any case, if MDNP can be understood from the VOC then there should be again red symbols showing that? Why do you show the effect of phenol constraints in the lower panels when phenol is not expected to contribute to the formation of DMNP and MDNP? In addition, the symbol style is chosen such, that overlapped curves cannot be seen very well.

And as already mentioned above, even if the model predicts the levels of the observations quite well, the time behavior does not really match.

Thank you for your comment. The revised Figure S3 is displayed in the supplementary information.

The previous overlapped model estimations are displayed in different panels to make it clear to readers. Besides, the reason why we showed the effect of phenol constraints to DMNP and MDNP

was that there were non-linear effects of oxidation capacities and radical concentration when phenol was constrained (line 216 - 218). As a result, there were indeed slight differences in estimating

DMNP and MDNP between these model scenarios (Figure S3). We agree that time behaviors showed discrepancies in some days. However, on the one hand, the trend of nitrated phenols agreed with the observations during heavy pollution episodes. On the other hand, the discrepancies between the model simulations and observations were regarded as primary emissions in this study. According to

NMF, NPs were also derived from primary emissions like biomass burning.

[Figure]

Figure S3. The measured concentration of nitrated phenols and their secondary formation simulation by the box model in different model scenarios.

Line 220-227: NO3 and OH contribute to C6H5O production. In the model phenol + NO3 and phenol + O have fixed branching ratios into C6H5O of 75% and 6%, respectively, and others of which about 80% lead to catechol in the OH case. Now I am wondering, does the green NO3 section comprise NO3-produced C6H5O or is it subsumed under the turquoise C6H5O part? For first case, how can the ratio of C6H5O path to catechol path vary since the phenol + OH reaction has a fix branching ratio? For the second case, assuming that NO3 will dominate C6H5O production the path to the other NO3 products seems to large. Please add a more detailed explanation what you used in detail to achieve the results in Figure 4 and Figure 5.

Thank you for your comment. The legend in Figure 4 caused misunderstanding and we have revised it accordingly. Neither $NO_3$ section comprised $NO_3$-produced $C_6H_5O$ nor it was subsumed under the turquoise $C_6H_5O$ part. The turquoise $C_6H_5O$ part was the **OH-phenol** reaction part which eventually formed $C_6H_5O$. We also revise the manuscript accordingly (line 221 - 231).

"Time series and diurnal profile of the loss of phenol during and without the heavy pollution episode were shown in Figure 4. It was obvious that the OH loss mainly took place during the day while $NO_3$ loss mainly happened at night. However, the fraction of these two pathways diverged dramatically taking the episode into account. During the heavy pollution episode, 46.7% of phenol lost from the pathway of OH-reaction which caused the production of phenoxy radical ($C_6H_5O$). We noticed that the $C_6H_5O$-$NO_2$ reaction was the only formation pathway of nitrophenol (Berndt and Böge, 2003). With the heavy pollution episode removed, the proportion of the $C_6H_5O$ production pathway of OH-reaction was only 5.4%. The phenol-OH reaction which produced catechol (then reacted with OH/$NO_3$, $NO_2$ to produce NC) was the predominant OH reaction (21.9%). The distinct pattern of the phenol-OH pathway which formed $C_6H_5O$ indicated a probable source of the nitrophenol accumulation during the heavy pollution episode. The high atmospheric reactivity and oxidation capacity in Beijing (Lu et al., 2019c; Yang et al., 2020) might be the foundation of high potential reactivity between phenol and OH radical".

The revised Figure 4 is displayed in the manuscript with clear descriptions in the legend. In addition, the branching ratios and rate constants of the box model were added to Figure 1.

[Figure]

(a) *heavy pollution episode*

[Figure]

(b) *with the episode removed*

**Figure 4.** Time series and the loss rate of phenol during the heavy pollution episode (a) and diurnal profile of the loss of phenol with the heavy pollution removed (b).

Line 228-234: I do understand what you wanted to say, but it is somewhat difficult to grab. It might be helpful to show the NP concentrations in Figure 5, too.

Thank you for your comment. The revised Figure 5 is displayed in the manuscript.

[Figure]

[Figure]

**Figure 5.** Time series of production and loss of nitrophenol (NP) during the heavy pollution episode (a) and diurnal profiles of production and loss of NP with the heavy pollution removed (b).

Line 241 – 244: What exactly is the Xylenol+NO2 reaction? The sentence starting with "As for

DMNP, the production …" is unclear. Please rephrase it. Where can I see the loss of DMNP in Figure

6?

Thank you for your comment. The revised sentence is "The production of DMNP increased rapidly from the xylenol-NO2 reaction during the daytime and decreased from noon" in line 244 - 245. The loss of DMNP (dilution) is displayed in Figure 6 in the revised version.

[Figure]

**Figure 6.** Production and loss of xylenol (a) and DMNP (b) during the sampling period.

In general, I would suggest, to modify the Figures remove overlap of elements. E.g., pie charts are partially in the Figure, partially outside. Formulas are crossing the frame of the diagrams, or in

Figure S1 the formulas are too large and overlap the MS peaks.

Thank you for your comment. We have double-checked the figures.

**Minor comments:**

line 94f: how can you be sure about the suggested structures? You used MS.

Thank you for your comment. The revised sentence is "The chemical structures of these NPs were identified by ToF-MS. The results of high-resolution peak fits of reagent ions and NPs could be found in Figure S1" in line 93-95.

We use several approaches to determine the molecular structure. First, the data processing procedures were conducted following previous studies (Priestley et al., 2018; Yuan et al., 2016). Second, we compare the structure with GC×GC-qMS data to further determine the structure and make sure the identification more reliable. For instance, the number of chemical structures of $C_6H_5NO_3$ in National

Institute of Standards and Technology (NIST) library is 15, nevertheless, only nitrophenol (NP) is probable in gas-phase samples in Beijing. This was guaranteed by non-targeted measurement of >50

gas-phase samples in autumn of Beijing utilizing thermal desorption comprehensive two-dimensional gas    chromatography-quadruple    mass    spectrometer    (TD-GC×GC-qMS).    The    campaign    was conducted from Sep. 1 to Oct. 31 in 2020. More than 3600 blobs were detected, including phenol, and isomers of NP, MNP, DMNP (Figure R1). The molecular weight of $C_6H_5NO_3$ (identified as NP in

CIMS), $C_7H_7NO_3$ (identified as MNP in CIMS), $C_8H_9NO_3$ (identified as MNP in CIMS)was 139,

153, and 167, respectively. The select ion chromatograms (SIC) of 139, 153, and 167 were displayed in Figure R2, R3 and R4. Despite NP, MNP, and DMNP, the molecular ion peaks of other compounds including these select ions were not 139, 153, and 167. This demonstrated that other structures of these molecular ion peaks occurred in the library of mass spectrums, however, they were not abundant in ambient air of Beijing. As a result, we identified seven peaks as nitrophenols in our study.

[Figure]

Figure R1. A typical chromatogram of gas-phase samples in Beijing analyzed by TD-GC×GC-qMS.

[Figure]

Figure R2. Select ion chromatogram ($C_6H_5NO_3$) of 139. Despite NP, the molecular ion peaks of eucapytol, naphthalenes, alkanes, and dibenzofuran were not 139.

[Figure]

Figure R3. Select ion chromatogram (C$_7$H$_7$NO$_3$) of 153. Despite MNP, the molecular ion peaks of other compounds were not 153.

[Figure]

Figure R4. Select ion chromatogram (C$_8$H$_9$NO$_3$) of 167. Despite MNP, the molecular ion peaks of other compounds were not 167.

line 100f: you calibrated with only one compound. Can you add something on the range of sensitivity expected for measurement of the addressed compounds by NO3-CIMS?

Thank you for your comment. Only one nitrophenol was used for calibration in this study, which could lead to uncertainty in quantifying other nitrophenols. We added uncertainty analysis in the SI

to make the reader more clear about how much the uncertainty is. Yuan et al. calibrated nitrophenol (NP), methylnitrophenol (MNP) and dinitrophenol (DNP) in the previous study utilizing nitrate-CIMS. The sensitivity of NP, MNP and DNP were 13.2, 16.6, 10.3 npcs $ppt^{-1}$, respectively (Yuan et al., 2016). The sensitivities of MNP and DNP ranged -26% and 22% from NP. Rebecca H.

Schwantes et al. estimated sensitivity factors for CIMS operated in both negative and positive mode using $CF_3O^-$ and $H_3O\ (H_2O)^+$. The estimated sensitivities of $o$-nitrophenol, 3-nitrocatechol,

4-methyl-2-nitrophenol were 1.48, 1.16 and 1.69, respectively. The sensitivities of NC and MNP

ranged 22% and -14% from NP (Schwantes et al., 2017). Even though uncertainties remain, we tend to believe that the addressed NPs calibrated by NP were correct in concentration levels and magnitudes. Besides, the secondary formation process simulated by the box model is constrained only by precursors of NPs measured by online GC-MS rather than the actual concentrations of NPs.

NMF model might be influenced by the uncertainties in the quantification. However, the high time resolution of CIMS increased sample inputs of the NMF model and reduced the uncertainties for this statistical approach. Even though the actual contrition of sources faces uncertainties, the proportion of source profiles is still reliable in this approach.

The text above was added to the drawing statement of Figure S2. In addition, we add uncertainty analysis in the manuscript (line 101 – 105) as follows, "The calibration curve was made by plotting the actual gas-phase NP concentration as the function of ion signals detected. The uncertainty in quantifying other NPs from the sensitivity of NP ranged from -26% to 22% (Schwantes et al., 2017;

Yuan et al., 2016). The addressed NPs calibrated by NP were correct in concentration levels and magnitudes. See more detail in Figure S2".

line 117: "other necessary packages", if the packages were necessary/important, you should name it otherwise I would skip that phrase.

Thank you for your comment. The revised sentence is "The data were analyzed by R 3.6.3 (R Core

Team, 2020), including packages of openair (Ropkins and Carslaw, 2012), Biobase (Huber et al.,

2015), NMF (Gaujoux and Seoighe, 2010), and ggplot2 (Wickham, 2016)" in line 120 – 121.

Figure S2: Why do you observe larger noise/fluctuations for the higher signals?

Thank you for your comment. The intensity of noise varies with the signal. The higher concentration/signal will increase the noise intensity accordingly. The signals were all normalized by reagent ions ($NO_3^-(HNO_3)_{0-2}$). The impact of fluctuations on calibration was reduced in this way.

**Typos etc:**

line 19f: contribution to production or concentration?

Thank you for your comment. We have modified the relevant content in the manuscript. The revised sentence is "Our results showed that secondary formation contributed 38%, 9%, 5%, 17% and almost

100% of the nitrophenol (NP), methyl-nitrophenol (MNP), dinitrophenol (DNP), methyl-dinitrophenol (MDNP or DNOC), and dimethyl-nitrophenol (DMNP) concentrations" in line

19 – 21.

line 34: "gained much concern", I would formulate that differently

Thank you for your comment. The revised sentence is "They are crucial species in forest decline" in line 34.

line 39: I believe that Beijing is still the capital …?

Thank you for your comment. We have modified the relevant content in the manuscript. The revised sentence is "Beijing is the capital city of China which retains a population of more than 20 million and more than 5 million private cars" in line 39 – 40.

line 40: "preserves … cars" , I would formulate that differently

Thank you for your comment. The revised sentence is "Beijing is the capital city of China which retains a population of more than 20 million and more than 5 million private cars" in line 39 – 40.

line 42: NAC is not defined

Thank you for your comment. The revised sentence is "Most of the studies in Beijing focus on particle-phase NPs (or so-called nitro-aromatic compounds, NACs)" in line 41 – 42.

line 45: either "spectrometry" or "spectrometers" (2x)

Thank you for your comment. We have modified the relevant content in the manuscript. The revised sentence is "Gas chromatography-mass spectrometers (GC-MS) and high-performance liquid chromatography-mass spectrometers (HPLC-MS) were commonly used to quantify the ambient concentration of NPs with accurate molecular information (Belloli et al., 1999; Harrison et al., 2005; Lüttke et al., 1997)" in line 45 - 47.

line 91: "…time resolution of the measurement…"? and 'The CIMS data processing was "conducted" by…' ?

Thank you for your comment. We have modified the relevant content in the manuscript in line 91 -93, as followed, "The original time resolution of the concentration of NPs was 1s. The CIMS data was processed by Tofware 3.0.3 (Tofwerk AG,  Aerodyne Research) in Igor Pro 7.08 (WaveMetrics Inc)".

line 112f: something is wrong with this sentence

Thank you for your comment. We have modified the relevant content in the manuscript. The revised sentence is "Totally 98 kinds of VOCs were measured, including alkanes, alkenes, aromatics, acetylene and oxygenated volatile organic compounds (OVOCs). The detailed information of these VOCs can be found elsewhere (Yu et al., 2021; Yuan et al., 2013)" in line 111 – 113.

line 119: I believe "functioned" is not the right word here.

Thank you for your comment. We have modified the relevant content in the manuscript. The revised sentence is "A zero-dimensional box model equipped with Master Chemical Mechanism (MCMv3.3.1) was utilized to simulate the secondary formation process of NPs" in line 115 – 116; and "The secondary formation process of NPs was simulated by a zero-dimensional box model equipped with Master Chemical Mechanism (MCMv3.3.1, http://mcm.leeds.ac.uk/MCM/home)" in line 123 – 125.

line 121: I would use present time (you should check the whole manuscript, there are more of these)

Thank you for your comment. We have revised the sentence into the present tense. The revised sentence "The related mechanism is presented in Figure 1" in line 124.

line 132: budget

Thank you for your comment. The revised sentence is "The budget analysis and the source apportionment were composed based on the constrained results of estimated phenol concentration by the ratio of phenol/CO" in line 138.

line 137: "total primary NPs 'were' calculated by subtracting", plural (you should check the whole manuscript, there are more like these)

Thank you for your comment. We have changed the word into "were". The revised sentence is "The total primary NPs were calculated by subtracting the secondary NPs from box model by the total NPs" in line 141 - 142. We have gone through the whole manuscript to check the words.

line 188: explanation for what exactly?

Thank you for your comment. The revised sentence is " The non-negligible secondary formation of nitrophenol from phenol oxidation was a plausible explanation for the higher concentration of DNP in Beijing" in line 193 - 194.

line 201f: something is wrong with this sentence

Thank you for your comment. We did not make this sentence clear. The revised sentences are as follows, "Nonetheless, NC and MNC (NPs with two -OH groups and one -NO$_2$ group) displayed a small peak at about 10:00 am, and revealed high concentrations at night. DNP and MDNP (NPs with one -OH groups and two -NO$_2$ groups) displayed distinct patterns from either NP or NC. DNP accumulated during the afternoon and began to decline after 5:00 p.m., suggesting that NO$_3$ oxidation of DNP might be a non-negligible sink. The diurnal profile of MDNP did not vary much during the whole day with a slight increase at night" (line 200 - 204).

line 244: is hailed the right word?

Thank you for your comment. The revised sentence is " DMNP mainly originated from the secondary formation process and its accumulation mainly took place in the afternoon while nitrophenol mainly occurred at night which were mainly derived from primary emission" in line 246 - 248.

Figure 1: scenarios (bold red)

Thank you for your comment. We have changed the word into "scenarios ".

Figure 8: I suggest, to use different colors for the grouping boxes

Thank you for your comment. We have revised Figure 8 accordingly.

[Figure]

**Figure 8.** Contribution of primary emission (in dark blue borderline) and second formation (in red borderline) of nitrated phenols. Primary emission was classified as biomass burning, coal combustion industry and vehicle exhaust which were resolved by non-negative matrix factorization (NMF). NPs in the legend referred to dinitrophenol (DNP), methyl-dinitrophenol (MDNP), methyl-nitrophenol (MNP), and nitrophenol (NP). Secondary formation of nitrophenol was categorized as benzene oxidation (<1%) and the oxidation of primarily emitted phenol (phenol oxidation, 37%). It was noticeable that nitrophenol derived from the primary phenol oxidation was much more important than the pathway from the traditional benzene oxidation in winter of Beijing.

Figure S3: see my major comments, I suggest, to improve the figure such that you can better separate the different cases.

Thank you for your comment. We have revised Figure S3 accordingly.

[Figure]

Figure S3. The measured concentration of nitrated phenols and their secondary formation simulation by the box model in different model scenarios.

Figure S8: I suggest, to use different colors for the grouping boxes

Thank you for your comment. We have revised Figure S8 accordingly.

[Figure]

Figure S8. Source profile from the PMF model. (a) Source profile of PMF results. SO₂, chloromethane, aromatics and 1,3-butadiene as the markers of coal combustion, biomass burning, industry and vehicle exhaust sources. (b) Contribution of primary emission (in dark blue borderline)

and second formation (in red borderline) of NPs.

[revised manuscript text omitted]

---

## Author Comment (AC2)

We thank the reviewers for their careful review of our manuscript. The comments and suggestions greatly improve our manuscript. Following is our point to point responses to the comments:

**Response to referee #2:**

This manuscript described the composition, variation, and sources of gas-phase nitrated phenols in Beijing during winter 2018. A box model was used to simulate the formation of nitrophenols. A NMF model was used to determine the primary sources of nitrophenols. Given the ubiquity of nitrophenols and the potentially important roles they play in influencing climate, this manuscript will be of interest to the atmospheric chemistry community. However, substantial revisions need to be made before this manuscript can be considered for publication.

We thank the reviewer for his careful review of our manuscript. Following is our point to point response to the comments.

1.  In general, I found the writing quality of the manuscript very poor. There were many parts of the manuscript where inappropriate words/terminology were used (e.g., "vicarious peaks" on line 238). There was also inconsistent use of tenses and punctuations. The poor writing made the manuscript very difficult (and frustrating) to read and understand. The writing has to be improved substantially. I strongly recommend the authors get someone with strong writing skills to help them improve the manuscript.

Thank you for your comment. We improve the writing substantially in the revised manuscript. In addition, we asked a native speaker to help us with the language editing.

2.  It was not clear from the manuscript whether calibrations were performed throughout the study or only at the beginning/end of the study. If calibrations were only performed at the beginning or end, how can the authors be sure that the sensitivity of their instrument was the same throughout the study?

Thank you for your comment. The calibrations were performed at the end of the campaign. The detailed information can be found in line 58 – 61 in the revised supplementary information. We agree that the sensitivity of CIMS might vary throughout the campaign. However, as the signals of nitrated phenols were all normalized by reagent ions ($NO_3^-( HNO_3)_{0-2}$), the fluctuations of sensitivity could be corrected in this way (Aljawhary et al., 2013; Duncianu et al., 2017).We added more description in the supplementary information. The details are as following:

[Figure]

Figure S2. (a) Background ions and ions detected during the calibration period (calibrated at the end of the campaign, on Jan 26, 2019); (b) Calibration line of ions ($y$) and the standard gas-phase concentration of nitrophenol ($x$). The signals were normalized by reagent ions ($NO_3^-( HNO_3)_{0-2}$).

Yuan et al. calibrated nitrophenol (NP), methylnitrophenol (MNP) and dinitrophenol (DNP) in the previous study utilizing nitrate-CIMS. The sensitivity of NP, MNP and DNP were 13.2, 16.6, 10.3

npcs ppt$^{-1}$, respectively (Yuan et al., 2016). The sensitivities of MNP and DNP ranged -26% and 22%

from NP. Rebecca H. Schwantes et al. estimated sensitivity factors for CIMS operated in both negative and positive mode using $CF_3O^-$ and $H_3O (H_2O)^+$. The estimated sensitivities of

$o$-nitrophenol, 3-nitrocatechol, 4-methyl-2-nitrophenol were 1.48, 1.16 and 1.69, respectively. The sensitivities of NC and MNP ranged 22% and -14% from NP (Schwantes et al., 2017). Even though uncertainties remain, the addressed NPs calibrated by NP were correct in concentration levels and magnitudes. Besides, the secondary formation process simulated by the box model is constrained only by precursors of NPs measured by online GC-MS rather than the actual concentrations of NPs.

NMF model might be influenced by the uncertainties in the quantification. However, the high time resolution of CIMS increased sample inputs of the NMF model and reduced the uncertainties for this statistical approach. Even though the actual contrition of sources faces uncertainties, the proportion of source profiles is still reliable in this approach.

3. Why was only one nitrophenol used for calibration? I don't think this is appropriate since different nitrophenolic compounds will have different CIMS sensitivities. Have the authors done other calibration tests to determine how the sensitivities of nitrophenolic compounds can differ? Uncertainties in the quantification of ambient nitrophenols may have contributed to the differences between their ambient observations and model predictions.

We agree with the reviewer. Only one nitrophenol was used for calibration in this study, which could lead to uncertainty in quantifing other nitrophenols. We added uncertainty analysis in the SI to make the reader more clear about how much the uncertainty is.

Yuan et al. calibrated nitrophenol (NP), methylnitrophenol (MNP) and dinitrophenol (DNP) in the previous study utilizing nitrate-CIMS. The sensitivity of NP, MNP and DNP were 13.2, 16.6, 10.3 npcs ppt$^{-1}$, respectively (Yuan et al., 2016). The sensitivities of MNP and DNP ranged -26% and 22% from NP. Rebecca H. Schwantes et al. estimated sensitivity factors for CIMS operated in both negative and positive mode using $CF_3O^-$ and $H_3O (H_2O)^+$. The estimated sensitivities of $o$-nitrophenol, 3-nitrocatechol, 4-methyl-2-nitrophenol were 1.48, 1.16 and 1.69, respectively. The sensitivities of NC and MNP ranged 22% and -14% from NP (Schwantes et al., 2017). Even though uncertainties remain, we tend to believe that the addressed NPs calibrated by NP were correct in concentration levels and magnitudes. Besides, the secondary formation process simulated by the box model is constrained only by precursors of NPs measured by online GC-MS rather than the actual concentrations of NPs. NMF model might be influenced by the uncertainties in the quantification. However, the high time resolution of CIMS increased sample inputs of the NMF model and reduced the uncertainties for this statistical approach. Even though the actual contrition of sources faces uncertainties, the proportion of source profiles is still reliable in this approach.

In addition, we add uncertainty analysis in the manuscript (line 103 – 104) as follows, "The uncertainty in quantifying other NPs from the sensitivity of NP ranged from -26% to 22%

(Schwantes et al., 2017; Yuan et al., 2016). The addressed NPs calibrated by NP were correct in concentration levels and magnitudes. See more detail in Figure S2". Figure S2 can be found as follows.

[Figure]

Figure S2. (a) Background ions and ions detected during the calibration period (calibrated at the end of the campaign, on Jan 26, 2019); (b) Calibration line of ions ($y$) and the standard gas-phase concentration of nitrophenol ($x$). The signals were normalized by reagent ions ($NO_3^-( HNO_3)_{0-2}$).

Yuan et al. calibrated nitrophenol (NP), methylnitrophenol (MNP) and dinitrophenol (DNP) in the previous study utilizing nitrate-CIMS. The sensitivity of NP, MNP and DNP were 13.2, 16.6, 10.3

npcs ppt$^{-1}$, respectively (Yuan et al., 2016). The sensitivities of MNP and DNP ranged -26% and 22%

from NP. Rebecca H. Schwantes et al. estimated sensitivity factors for CIMS operated in both negative and positive mode using $CF_3O^-$ and $H_3O (H_2O)^+$. The estimated sensitivities of

$o$-nitrophenol, 3-nitrocatechol, 4-methyl-2-nitrophenol were 1.48, 1.16 and 1.69, respectively. The sensitivities of NC and MNP ranged 22% and -14% from NP (Schwantes et al., 2017). Even though uncertainties remain, the addressed NPs calibrated by NP were correct in concentration levels and magnitudes. Besides, the secondary formation process simulated by the box model is constrained only by precursors of NPs measured by online GC-MS rather than the actual concentrations of NPs.

NMF model might be influenced by the uncertainties in the quantification. However, the high time resolution of CIMS increased sample inputs of the NMF model and reduced the uncertainties for this statistical approach. Even though the actual contrition of sources faces uncertainties, the proportion of source profiles is still reliable in this approach.

4.  How can the authors be sure that the seven peaks they tracked were nitrophenols? The MS

instrument only provides the m/z, not the molecular structure. Were nitrophenols also detected by the GCMS?

Thank you for your comment. The ToF-MS is excellent in identifying formulas of chemical compounds, not the molecular structure. However, we use several approaches to determine the molecular structure.

First, the data processing procedures were conducted following previous studies (Priestley et al.,

2018; Yuan et al., 2016). Second, we compare the structure with GC×GC-qMS data to further determine the structure and make sure the identification more reliable.

The listed nitrated phenols in the study were the most possible compounds for these molecular ion peaks. For instance, the number of chemical structures of $C_6H_5NO_3$ in National Institute of Standards and Technology (NIST) library is 15, nevertheless, only nitrophenol (NP) is probable in gas-phase samples in Beijing. This was guaranteed by non-targeted measurement of >50 gas-phase samples in autumn of Beijing utilizing thermal desorption comprehensive two-dimensional gas chromatography-quadruple mass spectrometer (TD-GC×GC-qMS). The campaign was conducted from Sep. 1 to Oct. 31 in 2020. More than 3600 blobs were detected, including phenol, and isomers of NP, MNP, DMNP (Figure R1). The molecular weight of $C_6H_5NO_3$ (identified as NP in CIMS),

$C_7H_7NO_3$ (identified as MNP in CIMS), $C_8H_9NO_3$ (identified as MNP in CIMS)was 139, 153, and

167, respectively. The select ion chromatograms (SIC) of 139, 153, and 167 were displayed in Figure

R2, R3 and R4. Despite NP, MNP, and DMNP, the molecular ion peaks of other compounds including these select ions were not 139, 153, and 167. This demonstrated that other structures of these molecular ion peaks occurred in the library of mass spectrums, however, they were not abundant in ambient air of Beijing. As a result, we identified seven peaks as nitrophenols in our study.

[Figure]

Figure R1. A typical chromatogram of gas-phase samples in Beijing analyzed by TD-GC×GC-qMS.

[Figure]

Figure R2. Select ion chromatogram ($C_6H_5NO_3$) of 139. Despite NP, the molecular ion peaks of eucapytol, naphthalenes, alkanes, and dibenzofuran were not 139.

[Figure]

Figure R3. Select ion chromatogram (C₇H₇NO₃) of 153. Despite MNP, the molecular ion peaks of other compounds were not 153.

[Figure]

Figure R4. Select ion chromatogram (C₈H₉NO₃) of 167. Despite MNP, the molecular ion peaks of other compounds were not 167.

5.  More information on the box model needs to be provided. For example, what branching ratios and rate constants were used in the model? Do the authors have any idea which reaction pathways are currently missing in their box model that may have contributed to differences between their ambient observations and model predictions?

Thank you for your comment. The branching ratios and rate constants of the box model were added to Figure 1 in the revised manuscript. Figure 1 can also be found as follows.

[Figure]

**Figure 1.** Mechanism related to the secondary formation of the nitrated phenols (NPs) in MCM 3.3.1

applied in this study. Different model scenarios differed in the constraints of the precursors. The basic model constrained the concentration of benzene by measurement from online GC-MS/FID. The other model scenarios constrained primary phenol concentration rather than benzene estimated by the ratio of phenol/NOy or phenol/CO from fresh vehicle exhaust.

The main missing reaction pathway in this study is gas-particle partitioning of NPs. According to Wang et al., the estimated proportions of gas-phase NP, MNP, and DMNP in Beijing were 99.2%, 94.9%, and <1%, respectively (Wang et al., 2019). Simulation of NP and MNP without gas-particle partitioning pathways faced small uncertainties as they mainly occurred in the gas-phase. The small proportion of DMNP in gas-phase and rather low concentration in particle-phase (0.55 ng m$^{-3}$, (Wang et al., 2019)) made the missing pathway not important. Meanwhile, gas-phase DMNP mainly came from secondary formation in this study and the concentration level of DMNP could be well explained by the box model.

We revised our manuscript as following (line 293 - 299):

The main missing reaction pathway in this study is gas-particle partitioning of NPs. According to Wang et al., the estimated proportions of gas-phase NP, MNP, and DMNP in Beijing were 99.2%, 94.9%, and <1%, respectively (Wang et al., 2019). Simulation of NP and MNP without gas-particle partitioning pathways faced small uncertainties as they mainly occurred in the gas-phase. The small proportion of DMNP in gas-phase and rather low concentration in particle-phase (0.55 ng m$^{-3}$, (Wang et al., 2019)) made the missing pathway not important. Meanwhile, gas-phase DMNP mainly came from secondary formation in this study and the concentration level of DMNP could be well explained by the box model. As a result, the missing pathway of gas-particle partitioning may not be important in this study.

[revised manuscript text omitted]

---

## Author Response (AR2)

We thank the reviewers for their careful review of our manuscript. The comments and suggestions greatly improve our manuscript. Following is our point to point responses to the comments:

Table 1 still needs a better layout. I suggest to use different line spacing to better group the data which belong together.

Response: Thank you for the comment. Table 1 has been revised as follows:

**Table 1. The concentration of phenol and nitrated phenols (NPs) in different sampling sites and their site categories, sampling time and analytical methods (ng m⁻³).**

| Sampling site | Site category | Sampling time | Method | phenol | NP | DNP | MNP | DMNP | NC | MDNP | MNC | References |
|---|---|---|---|---|---|---|---|---|---|---|---|---|
| Strasbourg area, France | urban and rural sites | annual mean | GC-MS | 0.4-58.7 | 0.01-2.2 | 5.6 | 2.6 | | | 0.1-0.3 [a] | | 1 |
| Rome, Italy | downtown | winter-spring | GC-MS | | 14.3 | | 13.9 | 2.0 (1.0) [b] | | | | 2 |
| Great Dun Fell, England | remote site | spring | GC-MS | 14-70 | 2-41 [c] | 0.1-8.5 | | | | 0.2-6.6 | | 3 |
| Beijing, China | regional site | spring | LC-MS | | *143-566* [d] | | *7.1-62* [e] | | *0.06-0.79* [f] | | *0.017* [g] | 4 |
| Milan, Italy | polluted urban site | summer | HPLC | 400 | 300 | | | | | | | 5 |
| northern Sweden | dairy farms | autumn-winter | TD-GC | 3000-50000 | | | | | | | | 6 |
| Manchester, UK | with Bonfire Plume Removed | autumn-winter | ToF-CIMS | | 780 | | 630 | | | | | 7 |
| Ottawa, Canada | selected dwellings sites | winter | TD-GC-MS | 10-1410 | | | | | | | | 8 |
| Santa Catarina, Brazil | near a coal-fired power station | winter | GC-FID | 980-1600 | | | | | | | | 9 |
| Switzerland | urban site | winter | GC-MS | 40 | 350 [h] | | 250 [i] | | | 50 [j] | | 10 |
| Manchester, UK | measured during the bonfire night | winter | ToF-CIMS | | 3700 | | 3600 | | | | | 7 |
| Detling, United Kingdom | rural site | winter | MOVI-HR ToF-CIMS | | 0.02 | 3 | 5 | | 2.5 | | 8.2 | 11 |
| Beijing, China (this study) | urban site | winter | ToF-CIMS | *63* [k] *1013* [l] | 606.3 (511.1) | 243.5 (339.6) | 203.5 (156.6) | 46.2 (32.6) | 22.1 (12.4) | 26.0 (25.8) | 10.4 (6.3) | |

The estimated concentrations were displayed in the *italic* script. Standard variations were displayed in brackets. Nitrated phenols investigated in this study referred to nitrophenol (NP), dinitrophenol (DNP), methyl-nitrophenol (MNP), dimethyl-nitrophenol (DMNP), nitrocatechol (NC), methyl-dinitrophenol (MDNP) and methyl-nitrocatechol (MNC).

**Symbols:** [a] gas+particle phase; [b] 2,6-Dimethyl-4-nitrophenol; [c] 2/4-Nitrophenol; [d] 4NP, estimated; [e] 2M4NP+3M4NP, estimated; [f] 4NC, estimated; [g] 3M6NC+3M5NC+4M5NC, estimated; [h] 2-Nitrophenol; [i] 3M2NP+4M2NP; [j] 2,4-Dinitro-6-methyl phenol; [k] estimated by 0.3NOy; [l] estimated by 0.4CO

**References:** [1] (Delhomme et al., 2010); [2] (Cecinato et al., 2005); [3] (Lüttke et al., 1997); [4] (Wang et al., 2019b); [5] (Belloli et al., 1999); [6] (Sunesson et al., 2001); [7] (Priestley et al., 2018); [8] (Zhu et al., 2005); [9] (Moreira Dos Santos et al., 2004); [10] (Leuenberger et al., 1988); [11] (Mohr et al., 2013).

line 93/94: I suggest to clarify: "The chemical formula compositions .... were detected by TOF-MS The chemical structures cannot be derived from simple MS, they were derived by aids of chromatography methods.

And: The results of high resolution peak fits of reagent ions and NPs can be found in Figure S1 in line 9.

Response: Thank you for the comment. The sentence was clarified as follows (line 92-93): "The chemical formula compositions of these NPs were detected by ToF-MS. See more detail in Figure S1".

line 225: "We noticed that the C6H5O-NO2 reaction was the only formation pathway of nitrophenol (Berndt and Böge, 2003)."

Did the authors really notice that, or do they want to remark, that Berndt and Böge suggested that? In the latter case the sentence needs to be rephrased.

Response: Thank you for the comment. The sentence was revised as follows (line 223-225): "During the heavy pollution episode, 46.7% of phenol lost from the pathway of OH-reaction which caused the production of phenoxy radical ($C_6H_5O$). $C_6H_5O$ then reacted with $NO_2$ and formed nitrophenol (Berndt and Böge, 2003)".